# p63 is a key regulator of iRHOM2 signalling in the keratinocyte stress response

Paola Arcidiacono[1], Catherine M. Webb[1], Matthew A. Brooke[1], Huiqing Zhou[2,3], Paul J. Delaney[1], Keat-Eng Ng[1], Diana C. Blaydon[1], Andrew Tinker[4], David P. Kelsell [1] & Anissa Chikh[1]

Hyperproliferative keratinocytes induced by trauma, hyperkeratosis and/or inflammation display molecular signatures similar to those of palmoplantar epidermis. Inherited gain-of-function mutations in *RHBDF2* (encoding iRHOM2) are associated with a hyperproliferative palmoplantar keratoderma and squamous oesophageal cancer syndrome (termed TOC). In contrast, genetic ablation of *rhbdf2* in mice leads to a thinning of the mammalian footpad, and reduces keratinocyte hyperproliferation and migration. Here, we report that iRHOM2 is a novel target gene of p63 and that both p63 and iRHOM2 differentially regulate cellular stress-associated signalling pathways in normal and hyperproliferative keratinocytes. We demonstrate that p63–iRHOM2 regulates cell survival and response to oxidative stress via modulation of SURVIVIN and Cytoglobin, respectively. Furthermore, the antioxidant compound Sulforaphane downregulates p63–iRHOM2 expression, leading to reduced proliferation, inflammation, survival and ROS production. These findings elucidate a novel p63-associated pathway that identifies iRHOM2 modulation as a potential therapeutic target to treat hyperproliferative skin disease and neoplasia.

[1] Blizard Institute, Barts and the London School of Medicine and Dentistry, Queen Mary University of London, London E1 2AT, UK. [2] Department of Human Genetics, Nijmegen Centre for Molecular Life Sciences, Radboud University Nijmegen Medical Centre, Nijmegen, The Netherlands. [3] Department of Molecular Developmental Biology, Radboud University, Nijmegen, The Netherlands. [4] The Heart Centre, William Harvey Research Institute, Barts and the London School of Medicine and Dentistry, Queen Mary University of London, Charterhouse Square, London EC1M 6BQ, UK. These authors contributed equally: David P. Kelsell, Anissa Chikh  Correspondence and requests for materials should be addressed to D.P.K. (email: d.p.kelsell@qmul.ac.uk) or to A.C. (email: a.chikh@qmul.ac.uk)

K eratinocyte hyperproliferation and inflammation are common to many skin disorders, from the more prevalent conditions such as psoriasis and atopic eczema to the rarer monogenic skin diseases which include the palmoplantar keratodermas (PPKs). The PPKs are characterised by different patterns of hyperproliferative thickening of the palms and soles, which are often painful[1,2]. Furthermore, PPK can also be associated with non-cutaneous conditions such as hearing loss, cardiomyopathy and oesophageal cancer[3,4]. For example, inherited dominant mutations in *RHBDF2*, the gene encoding iRHOM2, are the genetic basis of the inherited syndrome Tylosis (PPK) with Oesophageal Cancer (TOC, OMIM: 148500)[5].

iRHOM2 is a proteolytically inactive member of the seven transmembrane family of Rhomboid serine proteases[6]. iRHOM2 can control activation and trafficking of ADAM17 (also known as TACE; TNFα converting enzyme) from the endoplasmic reticulum to the Golgi and then to the cell surface[7–10]. ADAM17 is a membrane-anchored metalloprotease with a wide range of substrates including cytokines (TNFα), many receptors (IL-6R, TNF-R), growth factors (TGFα, AREG) and adhesion proteins[11]. The autosomal dominant TOC-associated missense mutations, located in the highly conserved cytoplasmic amino-terminal domain of iRHOM2, lead to increased ADAM17 activity and the 'shedding' of its associated substrates at the cell surface[9]. TOC keratinocytes have constitutively high levels of, for example, TGFα, AREG, IL-6R and IL-6. Thus, TOC-associated iRHOM2 mutations promote cell growth and migration in keratinocytes, which

exhibit a constitutive wound-healing phenotype and display similar features to the inflammatory skin disease psoriasis and epithelial cancer cell lines[5,12].

We have recently described[12] an important role for iRHOM2 in the regulation of the epithelial response to physical stress by identifying Keratin 16 (K16) as a novel interacting binding partner. TOC-associated iRHOM2 alters the dynamics of K16 regulation, including the hetero-dimerisation with its type II binding partner K6. In vitro depletion of iRHOM2 in TOC keratinocytes reduced K16 expression, proliferation and pro-inflammatory signalling. In contrast to the hyperproliferative TOC palmoplantar epidermis, *rhbdf2*$^{-/-}$ mice have a much thinner footpad epidermis compared to control mice. This striking cutaneous phenotype is associated with loss of K16 expression.

In this study, we show the presence of different signalling mechanisms between normal interfollicular and palmoplantar epidermis. The stressed palmoplantar epidermis mirrors disorders of keratinocyte hyperproliferation. To date, no studies have investigated the transcriptional regulation of iRHOM2. Here, we demonstrate that iRHOM2 is a direct molecular target of the transcription factor p63, the 'master regulator' of epithelial development[13,14]. Alteration of p63 expression is observed in oesophageal cancer, not only in carcinomas, but also in dysplasia[15,16]. *TP63* is expressed as multiple isoforms from alternative promoters with either an N-terminal transactivation (TA) domain or a dominant-negative (ΔN) domain and, in

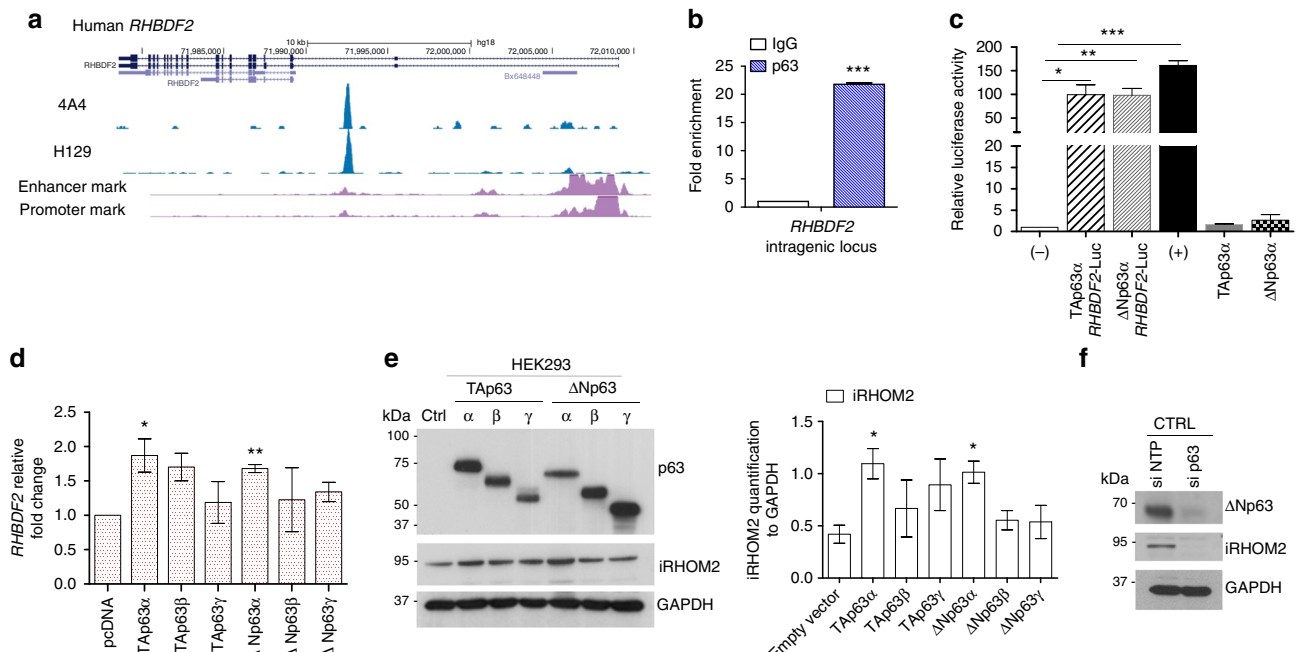

**Fig. 1** Identification of iRHOM2 as a p63 target gene in keratinocytes. **a** Screenshot of the UCSC genome browser from ChIP-seq analysis of normal human primary keratinocytes with two different antibodies (4A4, pan-p63 and H129, α-specific). The ChIP-seq study was previously reported. **b** Chromatin immunoprecipitation in control keratinocytes (CTRL) with anti-pan p63 (H137) and anti-IgG antibodies followed by quantitative PCR (ChIP-qPCR) analysis showed that p63 binds to *RHBDF2* intragenic region. Error bars represent SEM of three independent experiments and Student's two-tailed t-test value is shown; ***$p < 0.001$. **c** Luciferase assay for *RHBDF2* gene locus. The construct was transiently transfected into HEK293 cells in the absence (−) or in the presence of ΔNp63α or TAp63α. The (+) control corresponds to IRF6p229-Luc plasmid. The activity of the intragenic region was measured by luciferase assay and values are expressed relative to (−) set to 1. Data were analysed using two-tailed Student's t-test (*$p < 0.05$, **$p < 0.01$ and ***$p < 0.001$). **d** qRT-PCR for *RHBDF2* in HEK293 cells transfected with TA and ΔNp63 isoforms. The graph represents means and SEM of three biological replicates after 9 h of transfection. Statistical analysis was performed by Student's two-tailed t-test comparing pcDNA transfected cells to other samples. **e** Immunoblotting of HEK293 cells over-expressing p63 isoforms, showed TA and ΔNp63 isoforms (α-pan p63) and iRHOM2 expression after 9 h of transfection. GAPDH was used as loading control. The graph represents the means of the quantification, using ImageJ software, from three independent experiments relative to GAPDH. Error bars represent SEM and Student's two-tailed t-test values are given *$p < 0.05$. **f** Representative western blotting (WB) shows expression of ΔNp63 and iRHOM2 in normal keratinocytes (CTRL) treated with non-targeted protein (NTP) or p63 siRNAs. GAPDH was used as a loading control

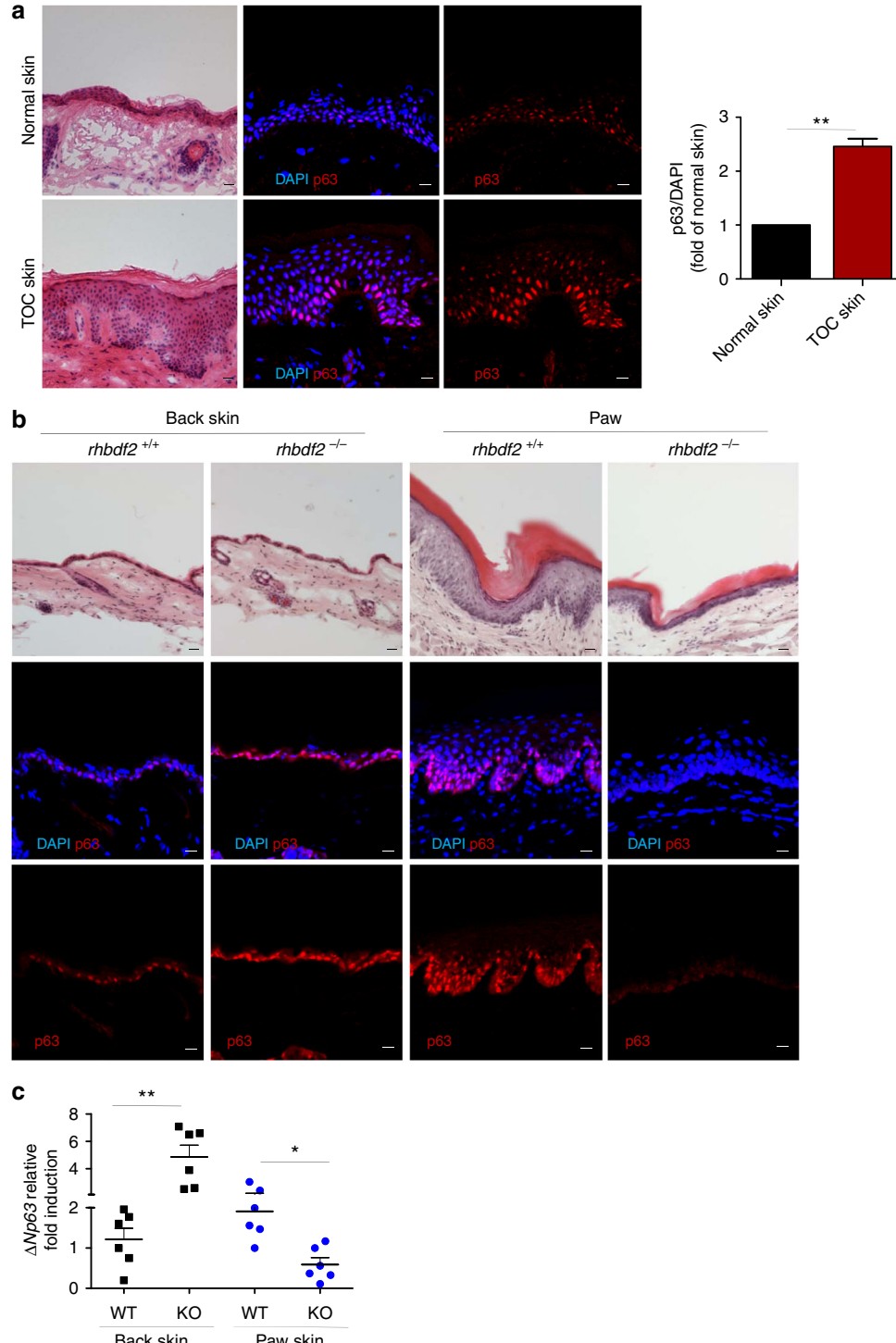

**Fig. 2** Distinct regulation of p63 in normal and hyperproliferative keratinocytes. **a** Representative images of H&E stained sections, displaying a hyperproliferative epidermis in TOC compared to normal interfollicular skin. Confocal microscopy analysis shows p63 (α-pan p63) expression in TOC and in control interfollicular skin. DAPI (blue) is used as a nuclear stain. Scale bars: 20 µm. Graph represents quantifications of p63 expression in ratio with the nuclei from three human samples. Student's two-tailed $t$-test value is shown, **$p < 0.01$. **b** Representative H&E stained back skin and fore-paw sections from 20-week-old $rhbdf2^{+/+}$ and $rhbdf2^{-/-}$ mice. Scale bar: 20 µm. Confocal analysis of p63 (α-pan p63) expression was performed in back skin and fore-paw sections of $rhbdf2^{+/+}$ and $rhbdf2^{-/-}$ mice. Scale bar: 20 µm. **c** qRT-PCR for $\Delta Np63$ from mRNA extracted from back skin and fore-paws of $rhbdf2^{+/+}$ and $rhbdf2^{-/-}$ mice. The graph shows $n = 6$ of each genotype. The statistical analysis was performed using Student's two-tailed $t$-test (*$p < 0.05$, **$p < 0.01$)

addition, these TAp63 and ΔNp63 transcripts can be alternatively spliced at the C-terminus to generate proteins designated α, β and γ[13]. The ΔNp63α isoform of p63 is expressed at high levels in the proliferative basal layer of the epidermis, suggesting the important role for this isoform in the biology of epithelial cells[13]. Recent studies have described that ΔNp63 overexpressing mice exhibit hyperproliferation, defects in terminal differentiation and an inflamed skin phenotype, demonstrating a key role of ΔNp63 in inflammatory skin disease[17,18].

We show that, in normal keratinocytes under physiological conditions, p63 positively regulates iRHOM2, while iRHOM2 antagonises ΔNp63 expression. In contrast, in hyperproliferative keratinocytes, there is an auto-regulatory feedback loop occurring between ΔNp63 and iRHOM2. We show that p63–iRHOM2 mediated signalling regulates ADAM17 activity and cellular functions including inflammation, proliferation, survival and oxidative defence. Furthermore, we identify SURVIVIN as a novel binding partner of iRHOM2 and as a p63 target gene. In addition, we also reveal a role of iRHOM2 in the epidermal oxidative defence response via its interaction with Cytoglobin (CYGB), a reported p63 target gene. Our findings implicate a novel signalling pathway involving p63 and iRHOM2 in the control of hyperproliferative skin diseases and squamous oesophageal cancer.

## Results

**p63 regulates iRHOM2 expression in normal keratinocytes**. To identify transcriptional regulators of iRHOM2, specifically if the *RHBDF2* gene encoding iRHOM2 may be a direct p63 target, we analysed an available p63 ChIP-seq data set performed in human and mouse keratinocytes[19,20]. This revealed that the human and mouse p63 binding sites are very well conserved at the intragenic region (Fig. 1a and Supplementary Fig. 1a). Furthermore, ChIP-qPCR confirmed that p63 binds to the *RHBDF2* gene locus (Fig. 1b). In addition, ChIP-qPCR analysis for the specific binding of p63 was performed with p21 (positive control), Thymidine kinase (TK, negative control) and a no-gene region (Chr11) (Supplementary Fig.1b). These data indicate that p63 binds to the intragenic enhancer region of *RHBDF2* and that iRHOM2 is a direct p63 target gene. In addition, we cloned this specific intragenic region, into the pGL3 enhancer plasmid[19], and tested whether the luciferase reporter gene activity is induced by TAp63α and ΔNp63α in HEK293 cells (human embryonic kidney cells). As shown in Fig. 1c, both TAp63α and ΔNp63α markedly upregulated luciferase activity. To further investigate whether p63 binding to iRHOM2 intragenic region affects iRHOM2 expression, each p63 isoform (ΔNp63α, ΔNp63β, ΔNp63γ, TAp63α, TAp63β and TAp63γ) was transiently transfected in HEK293 cells. qRT-PCR and western blot analysis showed that overexpression of the p63 isoforms modulates iRHOM2 expression, in particular TAp63α and ΔNp63α significantly induce iRHOM2 expression at mRNA and protein levels (Fig. 1d, e). Additionally, depletion of p63 in keratinocytes by a small interfering RNA (siRNA) which targets all p63 isoforms[21] significantly reduced endogenous iRHOM2 protein expression (Fig. 1f) as well as at the mRNA level (Supplementary Fig. 1c). Our data suggest that ΔNp63α, the major p63 isoform expressed in epidermis, is also the major p63 isoform that regulates iRHOM2 in keratinocytes.

To further investigate the expression of iRHOM2 in keratinocytes, HaCaT keratinocyte differentiation was induced in vitro by adding $Ca^{2+}$ to the culture medium. iRHOM2 expression was upregulated upon the induction of differentiation in a time-dependent manner (Supplementary Fig. 1d). ΔNp63, keratin 14 (K14) and involucrin expression was assessed by western blotting

to confirm that the keratinocytes were undergoing differentiation. As expected, ΔNp63 and K14, highly expressed under proliferative conditions, were reduced upon the induction of differentiation while involucrin expression was upregulated. Additionally, we also validated by qRT-PCR the increased expression of *RHBDF2* at mRNA level upon $Ca^{2+}$ shift (Supplementary Fig. 1e) and K14 was used as a control of the differentiation state. Moreover, iRHOM2 is expressed in the cytoplasm and plasma membrane of the basal and suprabasal layers of human epidermis (Supplementary Fig. 1f). Taken together, these results demonstrate that iRHOM2 is a transcriptional target of p63 and suggests that iRHOM2 might be implicated in the regulation of epidermal differentiation.

**Distinct regulation of p63 expression in keratinocytes**. Examination of the morphology of the stratified epithelia revealed a thicker epidermis in TOC compared with normal interfollicular skin as previously reported[9,22] (Fig. 2a). Immunohistochemistry showed increased p63 expression in the nuclei of the basal and suprabasal layers in TOC epidermis compared to control interfollicular skin (Fig. 2a). To better delineate the basal cells of the epidermis, we also performed immunofluorescence staining for Keratin 14 (K14), a marker of the basal layer (Supplementary Fig. 2a). Similarly, in the immortalised TOC keratinocytes, ΔNp63 was upregulated at the mRNA and protein levels when compared to control keratinocytes (Supplementary Fig. 2b and 2c). Microscopic examination of haematoxylin and eosin-stained skin sections showed no significant difference in the thickness of the *rhbdf2*−/− mice back skin compared to wild-type controls, while the *rhbdf2*−/− mice paw revealed a thinner epidermis compared to *rhbdf2*+/+ mice as published previously[12]. Back skin and footpads from *rhbdf2*−/− mice were immunostained for p63 and analysed by confocal microscopy (Fig. 2b). *rhbdf2*−/− mice showed increased p63 expression in the back skin but had reduced expression in their footpad epidermis compared to *rhbdf2*+/+ littermates (Fig. 2b). K14 was also used to delineate the basal cells of the epidermis (Supplementary Fig. 2d), as a 'bona fide' target of p63, interestingly, K14 follows the same p63 expression pattern. In addition, we also confirmed modulation of p63 expression by qRT-PCR and western blot analysis of extracts derived from paw and back skin of *rhbdf2*+/+ and *rhbdf2*−/− mice (Fig. 2c and Supplementary Fig. 2e). To investigate this apparent cell-context-dependent regulation of p63 by iRHOM2, we next used short hairpin RNA (shRNA) knockdown of iRHOM2, which was found to increase ΔNp63 protein expression in control keratinocytes, while sh-iRHOM2 TOC keratinocytes resulted in a downregulation of ΔNp63 expression (Supplementary Fig. 2f). These observations indicate that, in interfollicular skin, iRHOM2 represses ΔNp63 expression, whilst in hyperproliferative footpad skin and in TOC keratinocytes, iRHOM2 positively regulates ΔNp63 expression.

**iRHOM2–ADAM17 axis regulates p63 expression**. iRHOM2 regulates the maturation of the multi-substrate ectodomain sheddase enzyme ADAM17[7,8] and, in TOC derived keratinocytes, there is increased cleavage of ADAM17 substrates such as TNFα, IL-6R and EGFR ligands compared to control cells (9). TOC derived keratinocytes exhibit features of a constitutive 'wound-healing' phenotype in which the iRHOM2–ADAM17 axis plays a key role in skin barrier maintenance, inflammation and migration[5,9]. Here, the modulation of the iRHOM2–ADAM17 axis on p63 expression was investigated. Western blot analysis (Fig. 3a) revealed that ADAM17 depletion by siRNA led to an increase of ΔNp63 in control keratinocytes, whilst ΔNp63 was downregulated in TOC keratinocytes. These data were

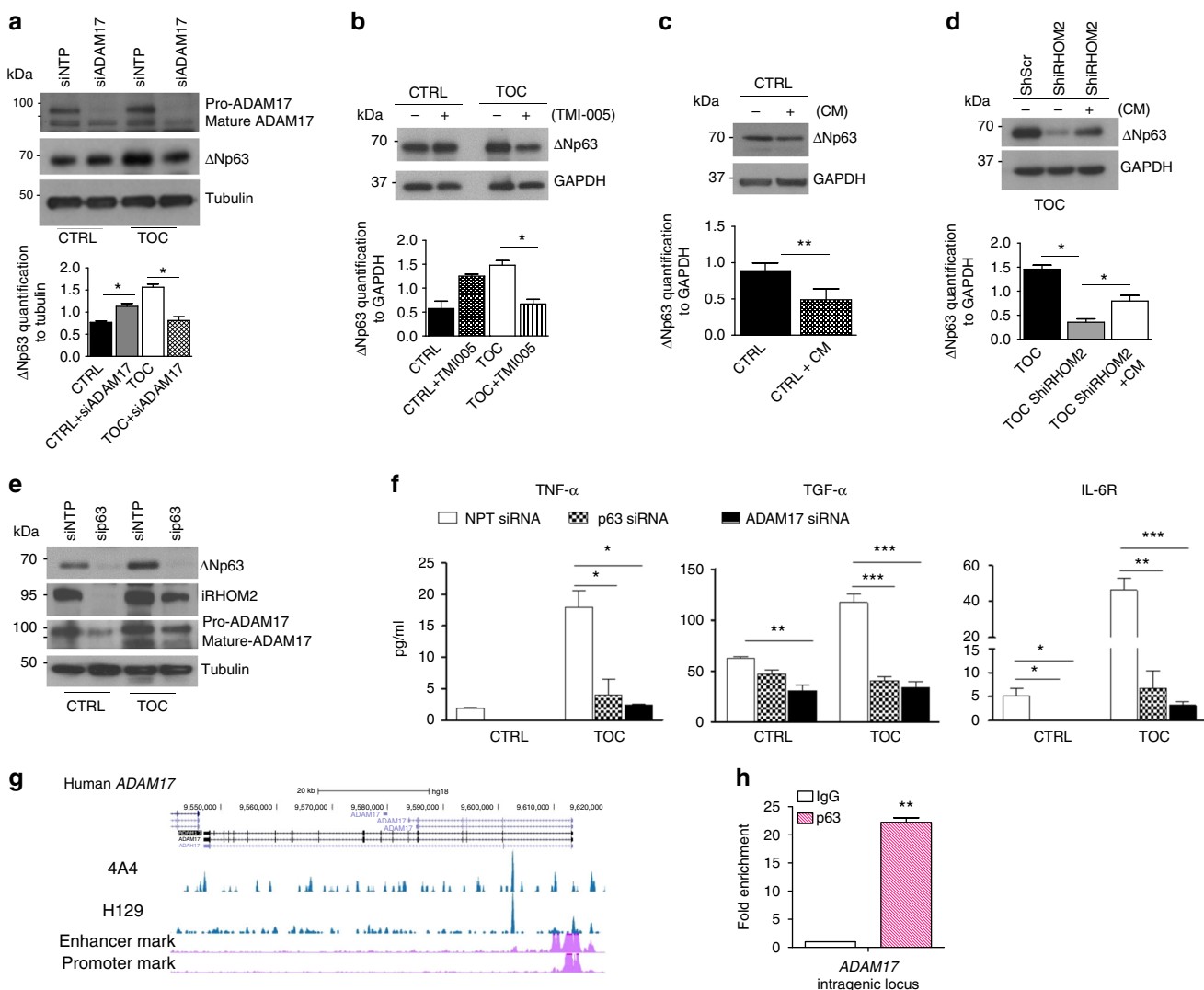

**Fig. 3** iRHOM2–ADAM17 axis regulates p63 expression. **a** Expression of ADAM17 and ΔNp63 by WB in control (CTRL) and TOC keratinocytes transfected with non-targeting pool (NTP) and ADAM17 siRNA. **b** Immunoblotting of ΔNp63 expression performed in CTRL and TOC keratinocytes treated with and without TMI-005 for 24 h. **c** Immunoblotting for ΔNp63 in CTRL keratinocytes and **d** in Sh Scr (Scramble) and Sh iRHOM2 TOC keratinocytes treated with or without conditioned media (CM) for 24 h. GAPDH was used as a loading control. **e** WB analysis for ΔNp63, iRHOM2 and ADAM17 in normal and TOC keratinocytes with NTP and p63 siRNA. TUBULIN was used as a loading control. All the quantifications included were performed by ImageJ software in comparison to the loading control in three independent experiments. Student's $t$ test was used, ($^*p < 0.05$ and $^{**}p < 0.01$). **f** Levels of TNFα, TGFα and IL-6R were assessed by ELISA from the supernatant of NTP, p63 and ADAM17 siRNAs in CTRL and TOC keratinocytes. Data are expressed as mean with SEM from four experiments. Statistical analysis was compared to NTP siRNA CTRL or TOC using one way ANOVA Dunnett's multiple comparison test ($^*p < 0.05$, $^{**}p < 0.01$ and $^{***}p < 0.001$). **g** Screenshot of the UCSC genome browser from ChIP-seq analysis of normal human primary keratinocytes with two different antibodies (4A4, pan-p63 and H129, α-specific). The ChIP-seq study was previously reported. **h** Chromatin immunoprecipitation in control keratinocytes (CTRL) with anti-pan p63 (H137) and anti-IgG antibodies followed by quantitative PCR (ChIP-qPCR) analysis showed that p63 binds to ADAM17 intragenic region. Error bars represent SEM of three independent experiments. For statistical evaluation Student's two-tailed $t$ test was used ($^{**}p < 0.01$)

consistent with our data above in iRHOM2 knock-down cells (Supplementary Fig. 2f). In agreement, inhibition of ADAM17 using the small molecule TMI-005 increased ΔNp63 in control keratinocytes but decreased its expression in TOC keratinocytes (Fig. 3b). As TOC-derived keratinocytes display an inflammatory phenotype[8,9], we next assessed whether the secretion of growth factors and cytokines in these cells may regulate the expression of ΔNp63.

Surprisingly, we found that conditioned media from TOC-derived keratinocytes reduced ΔNp63 expression (Fig. 3c) in control cells. As TNFα is known to induce keratinocyte differentiation and modulate p63 expression[23], keratinocytes treated with PMA (phorbol 12-myristate 13-acetate) showed downregulation of ΔNp63 expression in control cells (Supplementary Fig. 3a) confirming previous studies[24,25]. However, the addition of conditioned media from TOC keratinocytes to TOC keratinocytes with shRNA-mediated knockdown of iRHOM2 showed a restoration of ΔNp63 expression (Fig. 3d). These data suggested that 'the environment' of cell surface shed cytokines and growth factors can regulate ΔNp63 expression in a cell-context-dependent manner.

To examine this putative p63–iRHOM2–ADAM17 axis further, western blot analysis of both control and TOC keratinocytes with p63 siRNA knockdown (Fig. 3e) revealed

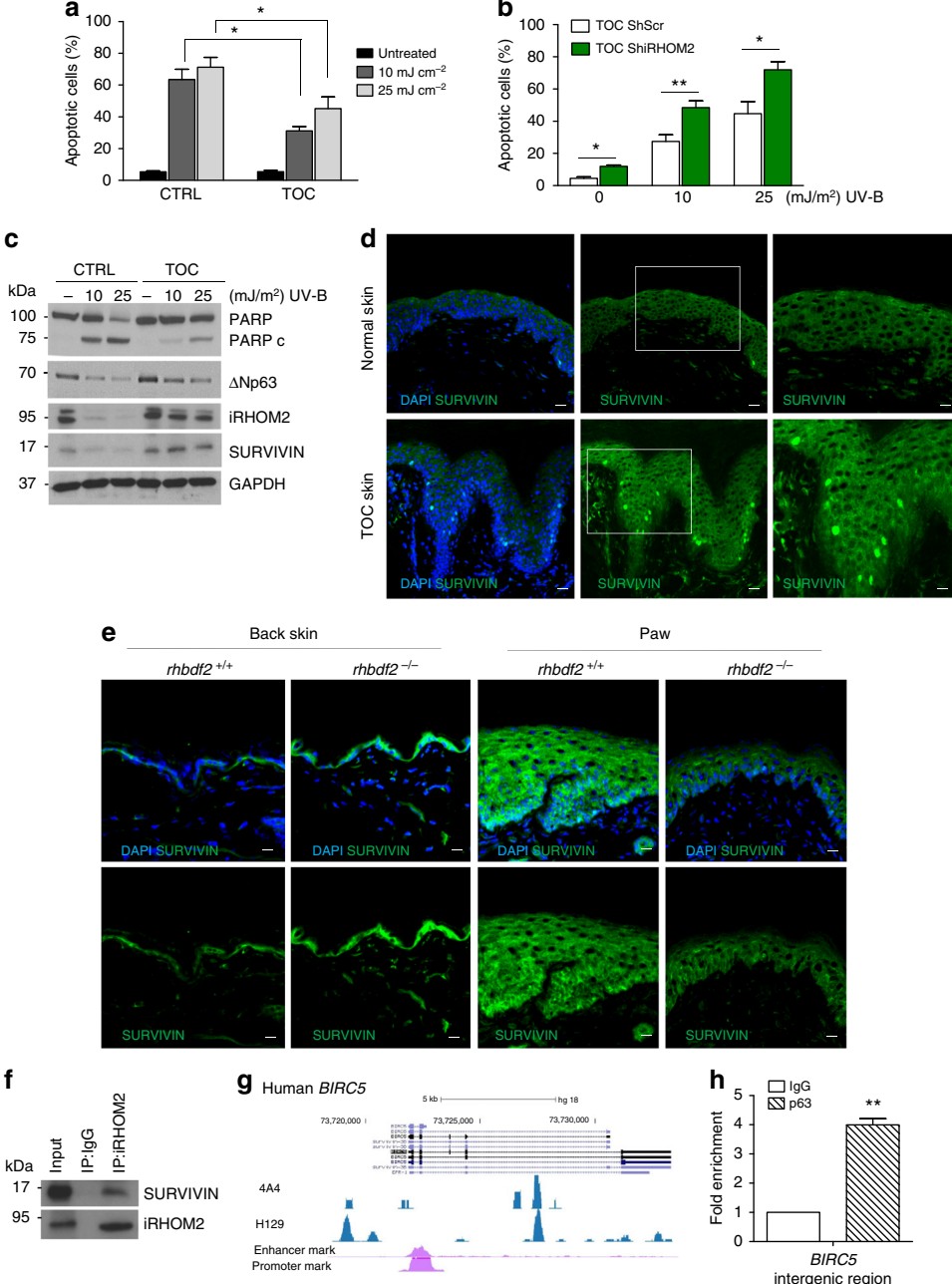

**Fig. 4** iRHOM2–p63 axis regulates apoptosis in keratinocytes by modulating SURVIVIN. **a** Percentage of apoptotic cells was quantified by using flow cytometric analysis of annexin-V-positive populations in control (CTRL) and TOC keratinocytes after 24 h of UV-B irradiation (10 or 25 mJ/cm²). Data represent means and SEM of four experiments. Statistical analysis was performed comparing TOC keratinocytes treated with 10 or 25 mJ/cm² and CTRL cells treated at the same doses using Student's two-tailed *t*-test (**p* < 0.05). **b** Percentage of apoptotic cells was quantified by using flow cytometric analysis of annexin-V-positive populations in Sh Scr and Sh iRHOM2-transfected TOC keratinocytes irradiated with 10 or 25 mJ/cm² for 24 h. Data represent mean with SEM of five independent experiments. For statistical evaluation Student's two-tailed *t*test was used (**p* < 0.05, ***p* < 0.01). **c** Immunoblotting for PARP, cleaved PARP (PARPc), ΔNp63, iRHOM2 and SURVIVIN in UVB irradiated CTRL and TOC keratinocytes with 10 or 25 mJ/cm² UV-B treatment. GAPDH is used as a loading control. **d** Representative confocal microscopy images from immunostaining of SURVIVIN in normal and TOC skin. DAPI (blue) is used as a nuclear stain. Scale bar: 20 μm. **e** Immunostaining of SURVIVIN was performed in back skin and fore-paw sections of *rhbdf2*⁺/⁺ and *rhbdf2*⁻/⁻ mice by confocal microscopy. DAPI (blue) is used as a nuclear stain. Scale bar: 20 μm. **f** CTRL keratinocyte lysates were immunoprecipitated using an anti-iRHOM2 antibody and immunoblotted with anti-SURVIVIN antibody. **g** Screenshot of the UCSC genome browser from ChIP-seq analysis of normal human primary keratinocytes with two different antibodies (4A4, pan-p63 and H129, α-specific) for BIRC5, gene encoding SURVIVIN. The ChIP-seq study was previously reported. **h** Chromatin immunoprecipitation in control keratinocytes (CTRL) with anti-pan p63 (H137) and anti-IgG antibodies followed by ChIP-qPCR analysis showed that p63 binds to the *BIRC5* intergenic region. Error bars represent SEM of three independent experiments. For statistical evaluation Student's two-tailed *t* test was used (***p* < 0.01)

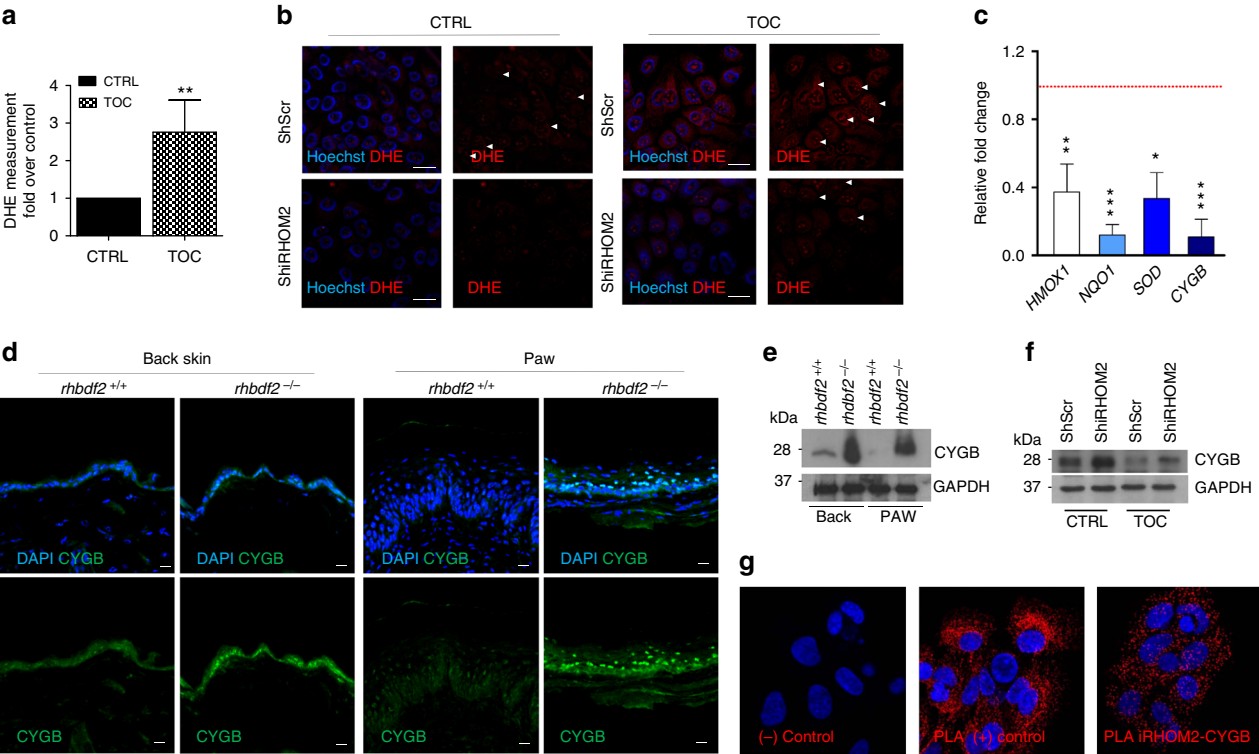

**Fig. 5** iRHOM2 regulates oxidative stress in hyperproliferative keratinocytes. **a** Quantification of dihydroethidium (DHE) staining in control (CTRL) and TOC keratinocytes by flow cytometry. Data are expressed as mean and SEM of three independent experiments. Statistical analysis was performed to compare CTRL and TOC cells using Student's two-tailed t-test (**p < 0.01). **b** DHE staining (red) in live Sh Scr or Sh iRHOM2-transfected CTRL and TOC keratinocytes by confocal microscopy. Hoechst 33342 (blue) was used to identify nuclei. **c** qRT-PCR of HMOX1, NQO1, SOD and CYGB in TOC keratinocytes was assessed as fold change compared to CTRL cell expression (dashed line). The graph represents means and SEM of three biological replicates. Statistical analysis was performed comparing CTRL and TOC keratinocytes using Student's two-tailed t-test (*p < 0.05, **p < 0.01, ***p < 0.001). **d** Immunostaining of CYGB was performed in fore-paw and back skin sections of rhbdf2[+/+] and rhbdf2[−/−] mice by confocal microscopy. DAPI (blue) is used as nuclear stain. Scale bar: 20 μm. **e** Representative WB for CYGB expression from proteins extracted from back skin and fore-paw of rhbdf2[+/+] and rhbdf2[−/−] mice. GAPDH was used as a loading control. **f** WB analysis of CYGB in Sh Scr and Sh iRHOM2-transfected CTRL and TOC keratinocytes. GAPDH was used as a loading control. **g** Representative confocal images from PLA experiments between iRHOM2 and CYGB in control keratinocytes. PLA between K6 and K16 was performed as a positive (+) control, while no primary antibodies were applied in negative (−) control

reduced expression of both iRHOM2 and ADAM17. As TOC keratinocytes are characterised by increased ADAM17 maturation and shedding of its substrates[9], we assessed the role of p63 in the regulation of ADAM17 protease activity. We demonstrated that depletion of p63 in both control and TOC keratinocytes resulted in decreased ADAM17 maturation and consequently, a reduction in the 'shedding' of TGFα, TNFα and IL-6R (Fig. 3f). To investigate the mechanism by which p63 regulates ADAM17 expression and its downstream pathway, we analysed the available p63 ChIP-seq data set performed in human and mouse keratinocytes[19,20]. This revealed that the human and mouse p63 binding sites are very well conserved at the intragenic region (Fig. 3g and Supplementary Fig. 3b). Furthermore, ChIP-qPCR confirmed that p63 binds to the ADAM17 gene locus (Fig. 3h). These data indicate that ADAM17 is a direct p63 target gene. To further investigate whether p63 binding to the ADAM17 intragenic region affects ADAM17 expression, p63 isoforms were transiently transfected in HEK293 cells, following which, qRT-PCR and western blot analysis showed that overexpression of TAp63α and ΔNp63α significantly induce ADAM17 at both mRNA and protein level (Supplementary Fig. 3c and 3d). Similarly, ADAM17 is also known to be a target of p73, another p53 homologue[26], as is p63. Thus, these results revealed that p63 can directly regulate both iRHOM2 and ADAM17 expression.

Additionally, in our previous study[12], we have shown that downregulation of iRHOM2 in both control and TOC keratinocytes was associated with reduced cell proliferation and migration. Here we found that depletion of p63 also reduced both proliferation and migration in control and TOC cell lines (Supplementary Fig. 3e and 3f).

**iRHOM2–p63 axis modulates resistance to apoptosis**. TOC epidermis displays improved barrier function, hyperproliferation and thickening of the palmoplantar regions[9,12]. As exposure to ultraviolet B (UV-B) light is an environmental stressor for basal keratinocytes, we investigated the cellular response in TOC keratinocytes. Upon UV-B treatment, TOC keratinocytes showed greater resistance to cell death compared to control keratinocytes as assessed by annexin-V staining using flow cytometry (Fig. 4a). Furthermore, control and TOC keratinocytes depleted for iRHOM2 exhibited an induction of apoptosis (Supplementary Fig. 4a). Depletion of iRHOM2 in TOC keratinocytes increased sensitivity to UV-B-induced apoptosis (Fig. 4b). Moreover, both cell lines depleted for p63 demonstrated induction of apoptosis following UV-B treatment (Supplementary Fig. 4b). These data suggest that iRHOM2 and p63 are involved in the regulation of the keratinocyte apoptotic pathway. Understanding the mechanism(s) through which TOC keratinocytes displayed

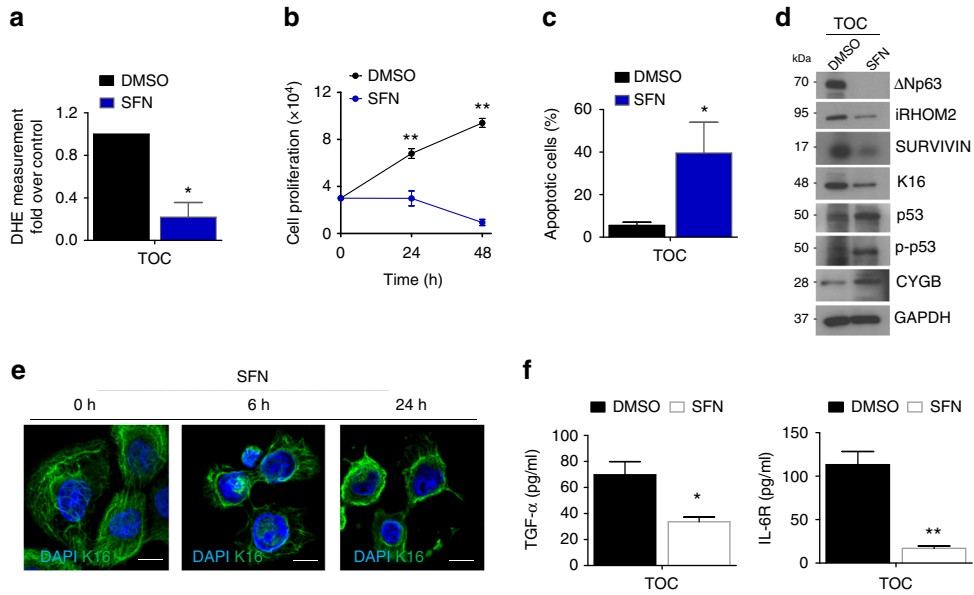

**Fig. 6** SFN represses iRHOM2–p63 pathway in TOC. **a** Quantification of DHE staining by flow cytometry in TOC keratinocytes treated with dimethyl sulfoxide (DMSO; the vehicle control) and sulforaphane (SFN) (10 μM) for 24 h. **b** Growth curves of TOC cells cultured with SFN (10 μM) or DMSO for 0, 24 and 48 h post treatment. Data are expressed as mean ± SEM. **c** Percentage of apoptotic cells was quantified by using flow cytometry analysis of annexin-V-positive populations detected after 24 h incubation with SFN (10 μM) or DMSO, in TOC cells. Bars represent mean values with SEM of four experiments. **d** Immunoblotting of lysates from TOC keratinocytes treated with SFN (10 μM) or DMSO for 24 h and analysed for ΔNp63, iRHOM2, SURVIVIN, K16, p53, phopho-p53 and CYGB expression. GAPDH is used as a loading control. **e** Immunofluorescence staining of K16 in TOC cells after 0, 6 and 24 h of SFN incubation. Scale bar: 20 μm. **f** ELISA for TGFα and IL-6R with the supernatants of TOC cells treated with SFN (10 μM) or DMSO for 24 h. All data are expressed as mean and SEM of three experiments. Statistical analysis was performed by Student's two-tailed $t$-test (*$p < 0.05$ and **$p < 0.01$)

resistance to cell death, the expression of inhibitors of apoptosis proteins (IAP) was investigated. Western blot analysis revealed an increased expression of SURVIVIN in TOC keratinocytes compared to control cells whilst expression of the other members of the IAP family was unchanged (Supplementary Fig. 4c). We also investigated the effect of UV-B stimulation on iRHOM2, ΔNp63 and SURVIVIN expression by western blot analysis. We first confirmed that the cells undergo apoptosis by assessing the expression of PARP, and observed lower levels of PARP cleavage in TOC keratinocytes after UV-B exposure in comparison to control cells. iRHOM2, ΔNp63 and SURVIVIN expression was virtually absent in UV-B exposed control keratinocytes but only slightly lower in TOC keratinocytes (Fig. 4c).

To explore this putative regulation of SURVIVIN by iRHOM2 further, confocal analysis revealed an increase in SURVIVIN expression in the cytoplasm and the nuclei of basal layer of keratinocytes in TOC epidermis, compared to control interfollicular skin (Fig. 4d). Immunostaining in $rhbdf2^{-/-}$ mouse skin showed increased levels of SURVIVIN expression in the back skin but reduced levels in their footpads in comparison to $rhbdf2^{+/+}$ littermate controls (Fig. 4e). In addition, we also confirmed the modulation of SURVIVIN expression by western blot analysis of protein extracts taken from back skin and paw of $rhbdf2^{+/+}$ and $rhbdf2^{-/-}$ mice (Supplementary Fig. 4d).

In agreement with data obtained from $rhbdf2^{-/-}$ mice, depletion of iRHOM2 resulted in an increase of SURVIVIN protein expression in control keratinocytes and a reduction in TOC keratinocytes (Supplementary Fig. 4e). To further explore the molecular mechanism by which iRHOM2 regulates SURVIVIN, we investigated whether the two proteins are associated in a complex. Co-immunoprecipitation analysis showed that endogenous iRHOM2 was able to efficiently immunoprecipitate SURVIVIN in control keratinocytes (Fig. 4f) and that endogenous SURVIVIN forms a complex with iRHOM2 (Supplementary

Fig. 4f). Taken together our data demonstrated that iRHOM2 is involved in the regulation of SURVIVIN.

As the depletion of p63 and iRHOM2 in both control and TOC keratinocytes induced apoptosis, we explored a possible regulation of SURVIVIN by p63. Prior studies have shown that SURVIVIN is negatively regulated by wild-type p53 but not mutated p53[27]. To identify transcriptional regulators of $BIRC5$, gene encoding SURVIVIN, we analysed the available p63 ChIP-seq data set performed in human and mouse keratinocytes[19,20]. This revealed that the human and mouse p63 binding sites are very well conserved at the intergenic region (Fig. 4g and Supplementary Fig. 4g). ChIP-qPCR revealed that p63 binds to $BIRC5$ gene locus (Fig. 4h). These data indicate that $BIRC5$ is a direct p63 target gene. To further investigate whether p63 binding to $BIRC5$ intergenic region affects SURVIVIN expression, p63 isoforms were transiently transfected in HEK293 cells. By qRT-PCR analysis, it was found that cells overexpressing TAp63α and ΔNp63α significantly induce $BIRC5$ at mRNA level (Supplementary Fig. 4h). Western blotting analysis confirmed the same results (Supplementary Fig. 4i). In addition, depletion of p63 by siRNA reduced endogenous expression of SURVIVIN in both control and TOC keratinocytes (Supplementary Fig. 4j). These findings support a model of reciprocal regulation between iRHOM2 and ΔNp63 in hyperproliferative keratinocytes that may play a role in the resistance to apoptosis via modulation of SURVIVIN.

**iRHOM2–p63 axis regulates oxidative stress.** Previous studies have shown that oxidative stress contributes to a form of palmoplantar keratoderma (pachyonychia congenita)[28], as well as inflammation[29], and may suppress apoptosis and promote proliferation[30]. Therefore, we investigated whether the iRHOM2 pathway could play a role in reactive oxygen species (ROS) regulation. We examined the production of ROS in the cells using

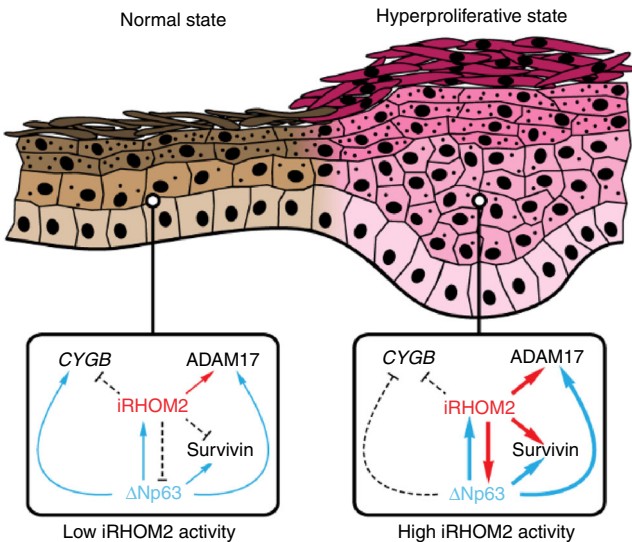

**Fig. 7** Proposed model. Model illustrating the regulation of iRHOM2–p63 pathway under normal and hyperproliferative states

dihydroethidium (DHE) dye by flow cytometry. In this assay ROS convert non-fluorescent DHE to fluorescent ethidium, which then intercalates into DNA. We found that TOC keratinocytes showed increased level of ROS compared to control cells (Fig. 5a). However, a reduction in DHE stained cells was observed in both control and TOC keratinocytes depleted for iRHOM2 (Fig. 5b). We also showed a significant decrease of ROS level in control and TOC keratinocytes depleted for iRHOM2 by flow cytometry (Supplementary Fig. 5a). Similarly, we also assessed the role of p63 in the regulation of ROS. An increase of ROS production in control keratinocytes depleted for p63 was observed while the levels were downregulated in TOC cells silenced for p63 (Supplementary Fig. 5b).

In order to explore a potential mechanism through which the p63–iRHOM2 axis may control oxidative stress in TOC keratinocytes, we evaluated the expression of genes associated with antioxidant pathways such as *NQO1* (NAD-(P)H:quinone oxidoreductases), *HMOX1* (Haem Oxygenase 1), *SOD1* (Superoxide Dismutase 1) and *CYGB* (Cytoglobin) by qRT-PCR. We observed a downregulation of these antioxidant genes in TOC keratinocytes compared to control (Fig. 5c) which is correlated with the increased levels of ROS in TOC keratinocytes. Interestingly, dysregulation of NRF2 was reported to contribute to palmoplantar keratoderma[28]. We investigated whether NRF2 may play a role in our pathway. Our data showed that *NRF2* is not modulated at mRNA levels in control and TOC keratinocytes (Supplementary Fig. 5c) as well as in the cells depleted for iRHOM2 (Supplementary Fig. 5d). Our studies then focussed on *CYGB* as it is a known p63 target gene[31] and ROS scavenger[32–34], and is also transcriptionally downregulated in TOC oesophagus[35,36]. Confocal analysis showed reduced CYGB expression in the basal layer of TOC epidermis compared to control interfollicular skin (Supplementary Fig. 5e). We also confirmed that p63 depletion led to a reduction of *CYGB* in control cells by qRT-PCR. However, in TOC cells, p63 siRNA resulted in an upregulation of *CYGB* expression (Supplementary Fig. 5f). These findings observed in TOC cells with p63 siRNA are correlated with the modulation of ROS observed in those cells.

To further support a role for iRHOM2 in CYGB regulation, an increase in Cygb expression was observed by confocal analysis in both back skin and footpad from *rhbdf2*[−/−] mice compared to *rhbdf2*[+/+] (Fig. 5d). In addition, we also confirmed modulation of

CYGB expression by western blot analysis of protein extracts derived from the back skin and paw of *rhbdf2*[+/+] and *rhbdf2*[−/−] mice (Fig. 5e). Similarly, shRNA knock-down of iRHOM2 showed an upregulation of CYGB in both control and TOC keratinocytes by western blot analysis (Fig. 5f). These data are correlated with the observed reduction of ROS production in iRHOM2-depleted cells and suggest a possible interaction may be occurring between iRHOM2 and CYGB. To investigate this possible interaction, proximity ligation assay (PLA) was performed in control keratinocytes and showed intense signals corresponding to formation of complexes between iRHOM2 and CYGB (Fig. 5g). Also, co-immunoprecipitation analysis showed that endogenous CYGB was able to efficiently immunoprecipitate iRHOM2 in control keratinocytes (Supplementary Fig.5g). These data indicate that iRHOM2 interacts with and represses CYGB, plus suggest both p63 and iRHOM2 participate in regulating the oxidative stress response.

**SFN suppresses p63–iRHOM2 pathway in TOC.** Oxidative stress has been linked previously to palmoplantar keratoderma associated with Keratin 16 (K16) mutations, and the use of SFN (Sulforaphane), a natural isothiocyanate compound found in cruciferous vegetables, rescued the palmoplantar keratoderma phenotype in *krt16*[−/−] mice[28]. Furthermore our recent study[12] has shown that *rhbdf2*[−/−] mice footpad showed a thinner epidermis, demonstrating that iRHOM2 regulates hyperproliferation and thickening of the palmoplantar epidermis. To explore whether SFN could affect TOC keratinocytes, cells were treated with this compound and analysed for ROS production by flow cytometry. Data showed that SFN treatment significantly reduced ROS production in TOC keratinocytes (Fig. 6a). Moreover, the treated TOC cells showed a significant reduction in cell proliferation (Fig. 6b) and were undergoing apoptosis (Fig. 6c). To investigate if SFN could regulate p63–iRHOM2 and associated downstream pathways, western blot analysis was performed and showed a downregulation of iRHOM2, ΔNp63 and SURVIVIN but an upregulation of CYGB following SFN treatment (Fig. 6d). In agreement with our previous report demonstrating that iRHOM2 regulates the stress-response keratin, K16[12], a downregulation of K16 expression (Fig. 6d) and the collapse of K16 filaments network upon SFN treatment in TOC keratinocytes was observed (Fig. 6e). It has been reported that apoptosis can result in keratin solubilisation, filament reorganisation and collapse[37]. In support of the positive feedback loop occurring between ΔNp63 and iRHOM2 in hyperproliferative keratinocytes, TOC keratinocytes silenced for p63 also demonstrated a downregulation of K16 (Supplementary Fig. 5h). Together, these data support the hypothesis that SFN inhibits the p63–iRHOM2 signalling pathway. SFN treatment reduced TOC-cell proliferation similarly to p63 (Supplementary Fig. 3e) or iRHOM2[12] downregulation. These cells also showed a reduction of ADAM17-mediated shedding of TGFα and IL-6R (Fig. 6f). Importantly, SFN treatment restored CYGB expression and reduced ROS levels. Moreover, SFN-treated cells also displayed a reduced SURVIVIN expression and were more sensitive to cell death. The increase in apoptosis was confirmed with the activation of p53 by phosphorylation of serine 15 (Fig. 6d).These findings indicated that SFN reduces the activation of p63–iRHOM2 pathway in TOC keratinocytes, resulting in reduced oxidative stress, inflammation, proliferation, and stress-response K16, and increased apoptosis.

## Discussion

Gain-of-function mutations in *RHBDF2*, the gene encoding iRHOM2, underlie Tylosis with Oesophageal Cancer (TOC)[5], a

syndrome characterised by palmoplantar epidermis thickening. We have recently shown the role of iRHOM2 in determining footpad thickness in humans and mice[12], with rhbdf2[−/−] mice displaying a thinner footpad epidermis, the opposite of the phenotype observed in human TOC palmoplantar epidermis. These data support an important role for iRHOM2 in regulating the epithelial response to stress. In this present study we provide new insights into the functional role of iRHOM2 in skin epidermal homoeostasis. Here, iRHOM2 has been identified as a new transcriptional target of p63 that is regulated in a cell-context-dependent manner. Considering that iRHOM2 is a key regulator of EGFR signalling and that iRHOM2 mutations cause an increase in the maturation and activity of ADAM17, we investigated its downstream pathway. Our data revealed that p63 has an impact on ADAM17 and its associated substrates, demonstrating a role of p63 in inflammatory skin diseases. In addition, we are also reporting that the iRHOM2–ADAM17 axis regulates p63 expression. These findings highlight a novel regulation linking iRHOM2 and p63 and suggest that some common pathways are occurring in keratinocyte homoeostasis. We have recently validated a model of study[12] in which we have established that mouse footpad skin and human TOC keratinocytes are considered to be a hyperproliferative model, compared to murine back skin and normal keratinocytes, which are equivalent to physiological conditions.

In the normal physiological state, such as in control keratinocytes or in mouse back skin, p63 positively regulates iRHOM2 while it antagonises ΔNp63 expression (Fig. 7). Thus, in this context, p63 depletion leads to a downregulation of its novel or known target genes such as iRHOM2, ADAM17, SURVIVIN and CYGB[31]. In contrast, iRHOM2 downregulation 'de-represses' p63 which in turn activates its downstream target genes such as SURVIVIN and CYGB. Our findings emphasise a critical role for the p63–iRHOM2 axis in normal skin. These data support previous findings showing significant roles for both p63[26,27,31] and iRHOM2[12] in regulating cellular proliferation, migration and inflammation.

In the hyperproliferative state, both TOC keratinocytes depleted for iRHOM2 and rhbdf2[−/−] footpad skin showed a downregulation of ΔNp63 expression. Our data highlight the fact that inflammation in TOC keratinocytes supports a hyperproliferative phenotype with ΔNp63 overexpression. These findings indicate that iRHOM2 positively regulates ΔNp63 expression in TOC. Consequently, we observed an increased expression of p63 target genes in hyperproliferative keratinocytes. Moreover, p63 depletion reduces iRHOM2 expression in TOC keratinocytes. Thus, this indicates that there is an auto-regulatory feedback loop occurring between iRHOM2 and ΔNp63 in the hyperproliferative state (Fig. 7), with similar regulation in proliferation, migration and inflammation.

Towards understanding the mechanisms through which TOC keratinocytes responded to stressors, we investigated how these cells respond to UVB treatment. This study allow us to identify SURVIVIN as a p63 target gene, which contributes to apoptosis resistance[38], confirming previous reports attributing a role in cell survival to ΔNp63 [39,40]. In addition, we have shown a key role for iRHOM2 in apoptosis via its direct regulation of SURVIVIN. Thus, in TOC keratinocytes, p63 siRNA or shRNA knock-down of iRHOM2 induces apoptosis and shows SURVIVIN downregulation. These observations confirmed previous studies demonstrating that SURVIVIN is often upregulated in cancer and dysplasia, driving resistance to cell death and contributing to progression of neoplasia[41].

Hyperproliferation and dysregulation of apoptosis are related to ROS production as it plays a role in these processes[30]. We have identified CYGB, a known p63 target gene[31], as a key modulator of ROS production in TOC keratinocytes. Prior studies have associated CYGB with cancer suppression[42] especially in oesophageal cells[36], and reported a transcriptional downregulation of CYGB in TOC. We confirmed low levels of CYGB expression in TOC and now identify CYGB as a novel interacting binding partner of iRHOM2. Thus, either the depletion of p63 or iRHOM2 in TOC keratinocytes, as well in rhbdf2[−/−] footpad skin, increases CYGB expression, which in turn dampens ROS release, indicating that the iRHOM2–p63 pathway influences maintenance of redox status.

In TOC keratinocytes, the natural antioxidant compound, sulforaphane (SFN) reduced ROS production and strikingly inhibited the iRHOM2–p63 pathway which can drive survival and inflammation. Previous studies have reported that SFN possesses potent chemopreventive efficacy in cancers[43] by inducing apoptosis[44] and NRF2 mediated induction of phase II detoxifying enzymes[45], protecting multiple organs from oxidative injuries[46] and reducing the inflammatory response[47]. Additionally, it has been well established that these different mechanisms act synergistically[48]. Consistent with a recent study, oxidative stress and dysfunctional NRF2 underlies the palmoplantar keratoderma disorder pachyonychia congenita associated with K16 mutations, with SFN treatment reducing the aberrant keratinisation in krt16 [−/−] mice[28]. Our findings demonstrate that SFN could also be used pharmacologically in TOC keratinocytes to target the downstream effects of the iRHOM2–p63 pathway, especially through CYGB induction. Thus, it is noteworthy to observe that SFN seems to mimic the depletion of p63 or iRHOM2 in TOC keratinocytes. As we have shown previously that rhbdf2[−/−] mice footpad showed a thinner epidermis[12] with absence of K16 expression, SFN treatment may have a similar impact on the hyperproliferative skin and oesophageal phenotype in TOC[28]. This study supports the model of the iRHOM2–p63 pathway regulating inflammation, hyperproliferation, oxidative stress and cell survival. Thus targeting of the iRHOM2–p63 axis in keratinisation disorders and dysplasia could have therapeutic potential.

## Methods

**Cell culture and reagents**. TOC cells are immortalised keratinocytes from a Tylosis patient carrying the UK RHBDF2 mutation have been described previously[5]. Control keratinocytes carrying the same immortalisation with human papilloma virus (HPV-16) open reading frames E6 and E7 as TOC cells. Cells were cultured in DMEM (Sigma), supplemented with 10% foetal bovine serum (FBS), 1% penicillin–streptomycin (pen–strep), 100 μM L-glutamine (Sigma) and keratinocyte growth supplement RM+ (RM+: containing EGF). HaCaT and HEK293 cells were cultured in DMEM supplemented with 10% FBS, 1% pen–strep and 100 μM L-glutamine. All cells were cultured in sterilised conditions at 37 °C with 5% of $CO_2$. RHBDF2 or negative scrambled shRNA for control and TOC keratinocytes were previously described[12].

**Antibodies**. The list of used antibodies is reported in Supplementary Information (Supplementary Table 1).

**Treatments**. For treatment (UV-B, Phorbol 12-myristate 13-acetate (PMA), condition media (CM), TMI005, $Ca^{2+}$ and Sulforaphane (SFN)), cultured cells were plated at the density of $2 \times 10^5$ cells in six-well plates and allowed to attach overnight at 37 ˚C, after 24 h, cells were treated. For UV-B, cells were irradiated at 10 and 25 mJ/cm². After UVB irradiation, the media was aspired and fresh culture medium was added (samples untreated were just changed with fresh medium). The cells were then harvested at 24 h post irradiation. For PMA (Sigma) treatment, the media was supplemented by the addition of 250 ng/ml PMA for 24 h. In condition media experiments, media from TOC keratinocytes, was centrifuged to eliminate any cells, then filtered using 0.2 μm filter (Millipore) and diluted in complete media (1:2). After these procedures the media was added to the cells for 24 h. For TMI005 (Apratastat, 1507 Axon Medchem UK) was dissolved in dimethyl sulfoxide (DMSO) and stock solution was freshly prepared and added to the cells at the final concentration of 500 nM for 24 h. The same DMSO concentration used to dilute the TMI005 was utilised as a negative control. To induce differentiation, calcium (2 mM) was added in the medium with 2% of FBS, when the cells until reaching ~90% confluence to induced differentiation. SFN was purchased from Sigma and

was dissolved in DMSO at the concentration of 40 mg/ml for the stock solution. SFN was added to cell cultures to obtain the final concentration of 10 µM, samples were analysed after 24 and 48 h post treatment.

**ChIP assay**. Chromatin was prepared from normal keratinocytes and immuno-precipitated with antipan-p63 (H137) overnight at 4 °C. ChIP assays were performed as previously described[21]. DNA extraction was carried out with phenol–chloroform. Purified DNA was diluted in water and subjected to qRT-PCR. Primer sequences used: ADAM17 F: 5′-CCTCACAATACTCAGCAAAA-3′, ADAM17 R: 5′-AGTCAGTAGGAGTGATTATG-3′; RHBDF2 F: 5′-TGTGCCCTTGCTTACCCCTG-3′, RHBDF2 R: 5′-CACTCATTTGCTCCTCCA-GAC-3′; BIRC5 F: 5′-CTCCTTTCCTGGTGCACCT-3′, BIRC5 R: 5′-CGGGGTGTGGTCCCTTTGGA-3′; TK F: 5′-GTGAACTTCCCGAGGCGCAA-3′, TK R: 5′-GCCCCTTTAAACTTGGTGGGC-3′; p21 F: 5′-ATGTA-TAGGAGCGAAGGTGCA-3′, p21 R: 5′-CCTCCTTTCTGTGCCTGAAACA-3′; Chr11 F: 5′-TTGCATATAAAGGAAACTGAAATGCT-3′, Chr11 R: 5′-TTACTGCCATGGGTCCGTATC-3′.

**Transfection**. HEK293 cells were transfected by using Lipofectamine 2000 reagent (Invitrogen) at a 1:2 ml/mg ratio with DNA, using 5 mg of plasmid DNA. The plasmids for p63 overexpression were a generous gift from Dr. Eleonora Candi.

**Gateway cloning and luciferase reporter assays**. To generate an RHBDF2 promoter reporter vector, recombinant plasmids were constructed using the Gateway cloning system (Invitrogen). A genomic fragment corresponding to the promoter region of RHBDF2 was PCR amplified using primers flanked by directional attB sites, after which this fragment was integrated into the pDONR221 gateway donor vector (Invitrogen) using BP clonase (Invitrogen).

The following, attB-flanked, primers were used for the amplification of the RHBDF2 promoter region Forward: GGGGACAAGTTTGTACAAAAAAGCAGGCTACCTGGAGGCTCACTCCAC TCT Reverse: GGGGACCACTTTGTACAAGAAAGCTGGGTTGGGATTACAGGCATGAG.

After transformation of OneShot TOP10 chemically competent E. coli (Thermo Fisher) and selection using kanamycin-containing LB medium, this vector was purified and analysed by restriction digestion using NaeI and HpaI restriction endonucleases. Entry clones with correct integration were then used to clone the promoter sequence into a modified pGL3-Enhancer vector (modified to contain attR integration sites) using LR clonase (invitrogen). Once again, these vectors were used to transform OneShot TOP10 competent E. coli, which were selected using LB medium containing ampicillin, purified, and analysed for correct integration by restriction digestion using EcoNI and NheI endonucleases. For luciferase reporter assays, HEK293 cells were plated at a density of $5 \times 10^4$ cells per well into 96-well plates. Twenty-four hours later, 50 ng of transient expression vectors encoding either TAp63α or ΔNp63α were transfected into these HEK293 cells either alone or alongside 50 ng of the firefly luciferase-expressing pGL3-RHBDF2 vector or the positive control IRF6_BS2_p229 vector, in addition to 5 ng of an internal control Renilla luciferase reporter vector, using lipofectamine transfection reagent. After 24 more hours, lysates were collected by treating with 20 µl passive lysis buffer per well for 15 min. Luciferase was assayed by the Dual-Luciferase system (Promega), with data presented as luciferase activity relative to untransfected cells.

**RNA interference**. For p63 knockdown, siRNA sequences against TP63 (ID 217143, ID 4893 and ID 217144 from Applied Biosystems) were used in combination to create a siRNA-p63 pool. For ADAM17 knockdown, siRNA sequences specific to against ADAM17 (L-003453-00-0005, OnTarget plus SMARTPool ADAM17 siRNA, Dharmacon Lafayette, USA). As a negative control, a non-targeting siRNA pool (D-001810-10-20, ON-TARGETplus, Dharmacon Lafayette, USA) was selected. Transfection was performed according to the manufacturer's protocol and optimised for a six-well plate. Normal and TOC keratinocytes were plated at 50% confluency and subjected to transfection on the following day using the transfection reagent Dharma FECT 1 (Thermo Fisher Dharmacon) and 60 nM final concentration of each siRNA. During siRNA application, antibiotics were removed from the cell culture medium. Transfection media were replaced with complete DMEM after 24 h. p63 and ADAM17 expression were assessed by western blot after 48 h.

**Immunoblot and co-immunoprecipitation**. After performing specific treatments, cells were harvested, washed three times with phosphate-buffered saline (PBS) and then lysed using lysis buffer [1 M Tris, 2.5 M NaCl, 10% glycerol, 0.5 M glycer-ophosphate, 1% Tween-20, 0.5% Nonidet P-40 and EDTA-free Complete Protease Inhibitor tablet (Roche)] for 15 min on ice. Protein concentration was measured using a Bradford Assay Kit (Bio-Rad). Equal amounts of protein were loaded and separated by sodium dodecyl sulphate-polyacrylamide gel electrophoresis (SDS-PAGE) (on 10% or 12% polyacrylamide gels) and transferred to a nitrocellulose membrane (Whatman). The blots were incubated with the specific antibodies and developed according to the manufacturer's instructions (ECL Immobilon Western,

Millipore). Antibodies were diluted in PBS containing 5% milk and 0.01% Tween 20. Co-immunoprecipitation experiments were performed as previously described[12]. Briefly, the lysates were collected after 30 min and pre-cleared by centrifugation at 13,000 rpm for 10 min at 4 °C. Protein concentration was calculated using the Bradford Protein Assay system (Bio-Rad). In total, 50 µl of Sepharose protein G (Amersham) were prepared according to manufacturer's instructions. IgG (Rabbit IgG sc-2027) was used as a control. 2 mg of protein lysate was added to beads and antibodies (2 µg) incubated overnight at 4 °C with rotation. The following day beads were washed three times for 5 min on ice in 1 ml of NP-40 wash buffer (50 mM Tris pH 8.0, 150 mM NaCl, 1 mM EDTA, 1% NP40) containing protease and phosphatase inhibitors (Roche). Beads were separated from antibody-protein complexes by boiling for 5 min in 2× Laemelli buffer and loaded onto a gel for SDS-PAGE electrophoresis. Uncut blots are supplied in Supplementary Figure 6.

**qRT-PCR**. Total RNA was prepared with RNeasy mini Kit (Qiagen). cDNA was synthesised using High-Capacity cDNA Reverse Transcription Kits strand (Thermo Scientific) for RT-PCR according to the manufacturer's instructions. Semi-quantitative PCR was carried out using Maxima SYBR Green/ROX qPCR master mix (Thermo Scientific, UK). Measurements were done in triplicate and normalised to levels of GAPDH mRNA for each reaction and analysed using the equation $n = 2^{-\Delta\Delta Ct}$. The following primers were used for qPCR analysis: for human genes GAPDH forward 5′-GGAGTCAACGGATTTGGTC-3′ and reverse 5′-GGCAACAATATCCACTTTACC-3′; ΔNp63 forward 5′-GAGTTCTGT-TATCTTCTTAG-3′ and reverse 5′-TGTTCTGCGCGTGGTCTG-3′;; HMOX1 forward 5′-AAAGTGCAAGATTCTGCCC-3′ and reverse 5′-GAGTGTAAG-GACCCATCGG-3′; NQO1 forward 5′-TCTATGCCATGAACTTCAATCC-3′ and reverse 5′-CTTCAGTTTACCTGTGATGTC-3′; SOD1 forward 5′-GGATGAA-GAGAGGCATGTTGGAGAC-3′ and reverse 5′-GTCTTTGTACTTTCTT-CATTTCCACC-3′; CYGB forward 5′-CTGTCGTGGA GAACCTGCAT-3′ and 5′-TGGAGTTAGGGGTCCTACGG-3′; RHBDF2 forward 5′-GGGCAAACTCA-GACTCGAAG-3′ and reverse 5′-CGCTGACTCCAAACCACTG-3′, ADAM17 forward 5′-GGTTCCTTTCGTGCTGGCGC-3′ and reverse 5′-AAGCTTCTC-GAGTCTCTGGTGGG-3′; NRF2 forward 5′-CAGCGACGGAAAGAGTATGA-3′ and reverse 5′-TGGGCAACCTGGGAGTAG-3′. For mouse genes ΔNp63 forward 5′-GTACCTGGAAAACAATGCCCAG-3′ and reverse 5′-CGCTATTCTGTGCCTGGTCTG-3′; GAPDH forward 5′-ACCA-CAGTCCATGCCATCAC-3′ and reverse 5′-TCCACCACCCTGTTGCTGTA-3′.

**The enzyme-linked immunosorbent assay (ELISA)**. After performing specific treatments, cell culture supernatants were used to detect TNF-α, IL-6R and TGF-α by ELISA using the human DuoSet ELISA kit as to the manufacturer's instructions (R&D System, UK). Results were expressed as means of four independent experiments with triplicate samples.

**Immunofluorescence**. Immunohistochemistry was performed on 5 mm frozen tissue or on cells plated in cover slips; sections were air-dried before processed. Cells/tissues were fixed in 4% paraformaldehyde (PFA) or in ice cold methanol–acetone (50:50 mixture) at room temperature for 15 min. If PFA fixation was used, samples were permeabilized with 0.1% Triton X-100. Cells/tissues were washed three times with PBS for 5 min each and incubated with 5% goat serum in PBS for 1 hour at room temperature to reduce nonspecific binding. After the cells/tissue were incubated with primary antibody in 5% goat serum overnight at 4 ˚C. The following day cells/tissues were washed three times with PBS and incubated with the secondary antibody conjugated with Alexa Fluor (Molecular Probes) in 5% goat serum for 1 h at room temperature. After three washes, cells/sections were incubated for 10 min with DAPI (100 ng/ml). Cells/tissues were mounted onto slides using Vectashield Mounting Medium (Vector Laboratories). Fluorescence was evaluated in one single plane by Zeiss 710 confocal microscopy (Carl Zeiss).

**Haematoxylin and eosin (H&E)**. Tissue sections were fixed in 4% PFA and stained with haematoxylin and eosin.

**Cell proliferation**. To analyse cell proliferation after performing transfection with siRNA, cells were plated at a seeding density of $1 \times 10^4$ per well and then harvested at 24 and 48 h. For SFN exposure, cells were seeded at $1 \times 10^4$ per well and then treated with SFN at 10 µM. Cells were counted after 24 and 48 h post treatment. Cell counts were performed using the Nucleocounter/Nucleocassette system (ChemoMetec, Denmark). Each experiment was repeated three times with triplicate samples.

**Scratch assay**. Cells were seeded into six-well plates in culture medium and were transfected with siRNA. The cells were incubated with mitomycin C (400 ng/ml, Roche) for 45 min to prevent proliferation. The scratch wound was created vertically to the centre of the well using a sterile tip to 200 µl. The cells were subsequently washed with PBS to remove the detached cells and then replenished with

fresh medium. The wound was monitored and photographed immediately (time 0) and after 24 h from the creation of the scratch using a phase-contrast microscope. The wound area was evaluated by using Image J Software. The percentage was calculated using the following equation: [Wound area (0h)−wound area (Xh)]×100/wound area (0h) = % wound closure.

**Annexin V assay**. Apoptosis was assessed by flow cytometry using FITC Annexin-V (Becton Dickinson) and DAPI. The cells were harvested by the addition of trypsin, centrifuged for 5 min at 1200 rpm, and washed with PBS. Cells were stained with FITC-Annexin V for 15 min and the DAPI (200 ng/ml) was added. Samples were analysed by a BD FACSCanto II Flow Cytometer (BD, UK). Data were expressed as means of total apoptosis, the sum of early and late apoptosis. The gating strategy is reported in Supplementary Fig.7.

**Proximity ligation assay**. PLA was performed using the Duolink in situ kit (Sigma) according to the manufacturer's instructions. Cells were plated on coverslips in 12-well plates and after 24 h fixed with methanol acetone or PFA. Following fixation, cells were incubated in Duolink blocking solution for 30 min at 37 °C. Primary antibodies were diluted in Duolink Antibody diluent and added to the cells overnight at 4 °C. The following day the cells were washed in Wash buffer A two times for 5 min. The PLA plus and minus probes were diluted in antibody diluent and added to the cells, then incubated for 1 h at 37 °C. The cells were washed in Wash buffer A two times for 5 min. The ligation-ligase was prepared according to instructions and applied to cells for 30 min at 37 °C. After washes, amplification with Duolink Amplification-Polymerase solution was performed for 100 min at 37 °C. The cells were washed in Wash Buffer B two times for 10 min and then mounted with Duolink in situ mounting medium with DAPI before visualisation with Zeiss 710 confocal microscopy (Carl Zeiss). As a positive control binding between K6 and K16 was analysed, while as a negative control no primary antibodies were applied. Quantifications were performed using Image J Software.

**Measurement of ROS production**. The quantification of intracellular ROS was based on the oxidation of dihydroethidium (DHE, Thermo Fisher). Briefly, normal and TOC keratinocytes ($2 \times 10^5$) were plated in each well of six-well plate. Cells were then washed twice with PBS and incubated with DHE (5 μM) at 37 °C for 30 min in the dark. After incubation, the cell were washed twice with PBS and were analysed by a BD FACSCanto II Flow Cytometer (BD, UK). Data represented the value of ROS in the live cells by a ratio referred to the control. For staining, normal and TOC keratinocytes ($5 \times 10^4$) were plated on cover slip in each well of 12-well plate. The cells on cover slip were washed twice with HBSS after exposure to DHE (Thermo Scientific) working concentration (5 μM) at 37 °C for 30 min in the dark. Before microscopy, Hoechst 33342 (0.1 μg/ml, Thermo Scientific) was added to stain the nuclei. The fluorescence was analysed using Zeiss 710 confocal microscopy (Carl Zeiss).

**Mouse studies**. $rhbdf2^{-/-}$ mice were generated as previously described[12]. The fore paw footpad epidermis from 20-week-old $rhbdf2^{-/-}$ or WT female mice were dissected and fixed in OCT on dry ice. Immunohistochemical staining was performed as previously described and visualised using the Zeiss 710 Confocal Microscope (Carl Zeiss). Biopsies from WT female mice ($n = 6$ per genotype, back and paw skin) were snap frozen, pulverised, and dissolved in TRIzol reagent for RNA preparation (Invitrogen) according to manufacturer's protocol. RNA samples were treated with RNase-free DNaseI (Qiagen). Western blot analysis of mouse skin extract were prepared in urea buffer 8 M (8 M urea, 1 M thiourea, 0.5% CHAPS, 50 mM DTT, 24 mM spermine).

**Study approval**. All experiments involving mice followed the UK Animal Welfare Act guidelines and were approved by the UK Home Office (PPL 70/7665). Human skin biopsies were approved by the research ethics committees of the National Health Service (08/H1102/73) and informed consent was obtained from participants.

**Statistical analysis**. All statistical analyses were performed with Prism 6 Software (GraphPad). Data were analysed by the unpaired/paired two-tailed Student's $t$ test and one way ANOVA Dunnett's multiple comparison test. All experiments were performed at least three times independently or more indicated in the legends. The data are expressed as the mean ± standard error of mean (SEM).

$*p < 0.05$, $**p < 0.01$ and $***p < 0.001$.

**Data availability**. The data that support the findings of this study are available from the corresponding authors (D.P.K., A.C.) upon reasonable request.

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

## Acknowledgements

We would like to thank Prof. Nick Reynolds and Dr. Richard Grose for critical reading of the manuscript. We are grateful to Dr. Eleonora Candi for providing constructs. We would like to thank Dr. Gary Warnes, Flow Cytometry Core Facility Manager for technical advice; Dr. Belén Martín-Martín and Dr. Jan Soetaert, Microscopy facility for assistance with the confocal microscopy and Rebecca Carroll (BMS2) core Pathology laboratory for technical support with tissue samples. The study was supported by grants awarded to D.P.K. from the Medical Research Council (MR/L010402/1), Cancer Research UK (C7570/A19107), the 2016 CHANEL-CERIES research award and the British Heart Foundation (RG/13/19/30568). The fellowship awarded to P.A. was from the European Commission and Regione UMBRIA under grant agreement FP7 Marie Curie Actions COFUND, I-MOVE (267232).

## Author contributions

P.A., D.P.K. and A.C. designed the research study. P.A., C.M.W., M.A.B. and A.C. performed the experiments and the analysis of the data. H.Z. provided the ChIP-seq data sets and their analysis. D.C.B. provided the human tissue slides. P.J.D., K.-E.N., A.T. provided the mouse samples. P.A., D.P.K. and A.C. wrote the manuscript.

## Additional information

**Competing interests:** The authors declare no competing interests.

