## [Peer Review File · Nature Communications]

Reviewer #1 (Remarks to the Author):

The paper by Arcidiacono and colleagues described iRHOM2 as direct p63 target genes. They claim that p63-iRHOM2 axis has a role in cell survival and anti-oxidant defence pathway through surviving and cytoglobin. Treatment of cells with the anti-oxidant compound sulforaphane down-regulated p63-iRHOM2 expression leading to reduced inflammation, apoptosis and ROS. They conclude that iRHOM2 could be a target for hyperproliferation and inflammatory skin diseases.

Overall, this paper is interesting and sheds new light on another molecular mechanism by which p63 regulates cell viability and inflammation. Nevertheless, not always the conclusion are supported by the data. Additional experiments and controls are required that will help to strengthen the Author's conclusions and improve the novelty, that in this present version is restricted to the identification of iRHOM2 as p63 target.

Specific points:

1. The data shown in Figure 1 demonstrating that p63 is upstream of iRHOM2 need to be strengthened. The major evidence for a direct link between p63 and iRHOM2 is a ChIP, that, in my opinion should be supported by additional experiments. Does ChipSeq data sets done in keratinocytes (many available both obtained from human and mice keratinocytes) confirm the data indicated in Fig1a and Fig 1b? Is the p63BS conserved among species. Does luciferase assay confirm a direct link between p63 and iRHOM2 RE? iRHOM2 decrease upon silencing of p63 in normal kerat, this could be also indirect effect. iRHOM2 accumulated at protein level during differentiation reaching high level when p63 is strongly down-regulated (day 5 calcium), it should be opposite, meaning that the target should mimic the pattern of expression of the inducer as for instance the p63 "bona fide" target K14. Importantly, Since they are describing a p63 transcriptional target gene, the experiments shown in HEK, HaCat and normal kerat, should be done also looking at mRNA level. The HEK experiments is confusing as DNp63 represses iRHOM2. Again the authors should look at mRNA first. The authors should include statistic analysis for the experiments shown.

2. Similar concern should be also addressed for p63-ADAM17 (Fig 3g) and p63-survivin (Fig 4i), showing a ChIP experiment only is not sufficient in my opinion.

3. Fig 2- IF staining shown is very poor quality. It is not possible to judge the tissue integrity since there is no DAPI staining. The tissues seem damaged in Fig2d "back skin" as p63 staining is not restricted in the nuclei but it is a "smear". The authors should also include additional markers to evaluate different layers of tissue sections.

4. Fig 4- statistical evaluation should be included for Fig4j. How the authors explain that there is not difference in apoptosis after 25mJ/cm²UV irradiation, Fig 4c? Again, DAPI staining is necessary to understand the structure of the tissue shown in Fig4f. In Fig 4j the quantification graph does not reflect the western blot shown. The authors should also include statistical evaluation. The authors should better demonstrate the survivin-iRHOM2 protein interaction (Fig 4h) by doing surviving IP and western blot for iRHOM2.

5. The data shown in Fig 5c is not clear, the authors should improve the staining. To demonstrate PLA specificity, the authors should also perform co-IP (Fig 5f).

Reviewer #2 (Remarks to the Author):

Overall, the data reported by Arcidiacono et al support their conclusions that iRHOM2 is a novel p63 target gene.

There are a few inconsistencies that the authors should address:

1. In Fig. 1e, iRHOM2 is expressed at high levels in HaCat cells have undergone Ca-induced differentiation, while p63 is essentially absent under these conditions (day 5). What regulates the high levels of iRHOM2 in the absence of p63?
2. In Sup. Fig 3a, the authors report that PMA treatment of keratinocytes in vitro reduces p63 expression. What happens in vivo since PMA treated mouse skin exhibits a profound hyperproliferative phenotype?
3. This statement "Co-immunoprecipitation analysis showed that endogenous survivin was able to efficiently immunoprecipitate iRHOM2 in control keratinocytes (Fig. 4h)." is incorrect, iRHOM2 was able to efficiently immunoprecipitate survivin.
4. Since sulforaphane is a well known inducer of Nrf2, which regulates numerous antioxidant response genes, can the authors exclude a role for Nrf2 in reduced ROS production? Are the same results obtained in Nrf2 knockout mice?

Reviewer #3 (Remarks to the Author):

In the manuscript by Arcidiacono et al. the authors show a complex interplay between the master epidermal transcription factor p63 and iRHOM2 that has implications for the keratinocyte stress response. Perhaps more interestingly, there exists a differential interplay between p63 and iRHOM2 that is dependent on normal vs hyperproliferative states and the anatomical location of keratinocytes. The authors further demonstrate that the p63-iRHOM2 mediated signaling axis regulates ADAM17 activity and in turn influences downstream cellular functions such as proliferation, survival and oxidative defense. To extend their studies, the authors focus on additional players, which are direct p63 targets and/or binding partner of iRHOM2 - this include survivin and cytoglobin (Cygb). Overall the manuscript is well-written and most of the conclusions are well-supported by experimental data. However, there are several issues that needs to be addressed to bolster the authors claims.

- 1) The p63 targets have been thoroughly identified by ChIP-seq experiments in both human and mouse keratinocytes. Yet, these published databases are not utilized for the identification of the specific targets that are chosen for the studies described in this paper. The examination of the proximal promoter region for the p63-binding sites is likely to be a rather limited and perhaps a misleading endeavor given that most of the p63 binding in the genome has been shown to be concentrated in the distal regulatory enhancers.
- 2) The ChIP data needs to be validated by qPCR to show quantitative enrichment of p63 binding.
- 3) The p63 overexpression studies in the HEK293 cells (Fig 1) and its effects on endogenous iRHOM2 expression is not convincing.
- 4) Since in both mouse and human keratinocytes, under most conditions, the Δ Np63 isoforms are overwhelmingly predominant, examination of TAp63 does not add much to the manuscript (Supp Fig 2 for e.g.)
- 5) The Immunofluorescence studies of human and mouse skin samples will benefit from additional markers (K14 for e.g.) to better delineate the basal cells and the rest of the epidermis. Additionally, H&E stains will also allow better appreciation of the morphology in these skin sections.
- 6) The authors have used normal and TOC keratinocytes – yet the details about these cell lines are quite sparse. Presumably they are immortalized cell lines – what passages were used?
- 7) What is the status of the epidermal differentiation program in the human TOC skin and in the iRHOM2-null mice? This aspect has been neglected in the current study.
- 8) The authors have interesting data that led them to posit that "in interfollicular skin, iRHOM2 represses Δ Np63 expression, whilst in hyperproliferative footpad skin and in TOC keratinocytes, iRHOM2 positively regulates Δ Np63 expression". There are several caveats to this assumption, principle being that these observations are merely correlative. In particular, the p63 immunofluorescence studies on the mouse samples as presented, does not really provide much

support for the authors conclusions and lacks a much-required quantitative evaluation. Since this is really a novel finding, it would be worth-while for the authors to shore up these observations by independent experiments (Western blot of skin extracts). Also, the authors state that the irhom2 -/- mice footpad phenotype displays a thinner epidermis, the opposite of the phenotype observed in TOC palmoplantar epidermis. This is not easily discernible in the Fig 2D panel. Again, a representative histology panel will be useful.

9) Statistical analysis is not adequate.

10) Finally, the manuscript as it stands now seems more of an attempt to bring together a few players that fit into the p63-iRHOM2 axis but falls short in breaking new ground in terms of either p63 or iRHOM2 biology. The limited studies performed on the mouse model of irhom2KO adds further confusion to the story line and the authors do not address the mechanistic basis of the footpad phenotype and its relationship to TOC states.

Replies to Reviewers:

Reviewers' comments:

Reviewer #1 (Remarks to the Author):

The paper by Arcidiacono and colleagues described iRHOM2 as direct p63 target genes. They claim that p63-iRHOM2 axis has a role in cell survival and anti-oxidant defence pathway through surviving and cytoglobin. Treatment of cells with the anti-oxidant compound sulforaphane down-regulated p63-iRHOM2 expression leading to reduced inflammation, apoptosis and ROS. They conclude that iRHOM2 could be a target for hyperproliferation and inflammatory skin diseases.

Overall, this paper is interesting and sheds new light on another molecular mechanism by which p63 regulates cell viability and inflammation. Nevertheless, not always the conclusion are supported by the data. Additional experiments and controls are required that will help to strengthen the Author's conclusions and improve the novelty, that in this present version is restricted to the identification of iRHOM2 as p63 target.

Reply:

We thank the reviewer for his/her encouraging comments about the importance and novelty of our findings linking p63 to iRHOM2 in keratinocyte stress response. We also appreciate his/her constructive criticism and useful suggestions which we hope we have fully addressed in this revised version of the manuscript as we detail point-by-point here below.

Specific points:

1. The data shown in Figure 1 demonstrating that p63 is upstream of iRHOM2 need to be strengthened. The major evidence for a direct link between p63 and iRHOM2 is a ChIP, that, in my opinion should be supported by additional experiments. Does ChIP-seq data sets done in keratinocytes (many available both obtained from human and mice keratinocytes) confirm the data indicated in Fig1a and Fig 1b? Is the p63BS conserved among species? Does luciferase assay confirm a direct link between p63 and iRHOM2 RE? iRHOM2 decrease upon silencing of p63 in normal keratinocytes, this could be also indirect effect. iRHOM2 accumulated at protein level during differentiation reaching high level when p63 is strongly down-regulated (day 5 calcium), it should be opposite, meaning that the target should mimic the pattern of expression of the inducer as for instance the p63 "bona fide" target K14. Importantly, since they are describing a p63 transcriptional target gene, the experiments shown in HEK, HaCat and normal keratinocytes, should be done also looking at mRNA level. The HEK experiments is confusing as DNp63 represses iRHOM2. Again the authors should look at mRNA first. The authors should include statistical analysis for the experiments shown.

Reply:

To further demonstrate that iRHOM2 is a p63 target gene, as suggested by the reviewer, we have validated using the available ChIP-seq data sets performed in human and mouse keratinocytes (Kouwenhoven et al., PLoS genetics 2010; Sethi et al., Nucleic Acid Res 2016). From these datasets, we confirmed that both the human and mouse p63 ChIP-seq sites are very well conserved (Fig.1a and Supplementary Fig.1a). We also performed ChIP-qPCR experiments and validated that p63 binds to the RHBDF2 intragenic region (Fig.2b) and only weakly to the RHBDF2 promoter. Furthermore, we confirmed by luciferase assay that the

RHBDF2 enhancer activity was positively regulated by Δ Np63 α and TAp63 α in HEK293 cells (Fig.1c).

We also understand the concern of the reviewer regarding the overexpression experiments and the confusion generated with the regulation of iRHOM2 upon calcium shift. We have now repeated the transient transfection experiments after optimizing the time of transfection to 9 hours for transactivation. This is in line with previous studies described for example in Candi et al., 2006 (JCS), where p63 isoforms could be detectable upon 3 hours post transfection. We observed that TAp63 α and Δ Np63 α significantly induce iRHOM2 expression at mRNA and protein levels (Fig.1d/e).

During keratinocyte differentiation, the reviewer mentioned that: “iRHOM2 accumulated at protein level during differentiation reaching high level when p63 is strongly down-regulated (day 5 calcium), it should be opposite, meaning that the target should mimic the pattern of expression of the inducer as for instance the p63 “bona fide” target K14”. Our findings may appear to be in contrast with other studies. In fact, the majority of the Δ Np63 target genes followed this principle, however it has been also reported that some genes do not respect this rule. For instance it has been described that ZNF750 is involved in epidermal differentiation, and shown to be upregulated during epidermal differentiation by Δ Np63 α which binds to its promoter (Sen et al., Dev Cell 2012). Our data suggest that Δ Np63 α , the major p63 isoform expressed in epidermis, is also the major p63 isoform that regulates iRHOM2 in keratinocytes.

Additionally, we now demonstrate that during keratinocytes differentiation, similar to ZNF750, iRHOM2 expression is increasing at both mRNA and at protein levels (Supplementary Fig.1d and 1e) mirroring what is seen in normal skin where iRHOM2 is expressed in the cytoplasm and plasma membrane of the basal and suprabasal layers of the epidermis.

2. Similar concern should be also addressed for p63-ADAM17 (Fig.3g) and p63-survivin (Fig 4i), showing a ChIP experiment only is not sufficient in my opinion.

Reply:

We have checked ChIP-seq data derived from human and mouse keratinocytes (Kouwenhoven et al., PLoS genetics 2010; Sethi et al., Nucleic Acid Res 2016) additionally for ADAM17 regulation (Fig.3g and Supplementary Fig.3b) and BIRC5 (Fig.4g and Supplementary Fig.4b). ChIP-qPCR experiments were performed and revealed that p63 binds to ADAM17 (Fig.3h) and BIRC5 (Fig.4h) within the analysed region suggesting that these two genes are also direct p63 target genes.

In addition, we also demonstrate increases expression of ADAM17 and BIRC5 at both mRNA and at protein level by overexpressing TA and Δ Np63 isoforms in HEK293 cells (Supplementary Fig.3c/d; Supplementary Fig.4h/i).

3. Fig 2- IF staining shown is very poor quality. It is not possible to judge the tissue integrity since there is no DAPI staining. The tissues seem damaged in Fig2d “back skin” as p63 staining is not restricted in the nuclei but it is a “smear”. The authors should also include additional markers to evaluate different layers of tissue sections.

Reply:

We have stained new sections and we have now updated Fig.2a/d. In the new image p63 is clearly seen in the nuclei and we have included the DAPI staining. The quality and resolution of the images presented have also been improved. As suggested by the reviewer we have performed additional staining for K14 in human and mouse to better delineate the basal cells of the epidermis (Supplementary Fig.2a/d).

4. Fig 4- statistical evaluation should be included for Fig4j. How the authors explain that there is not difference in apoptosis after 25mJ/cm²UV irradiation, Fig 4c? Again, DAPI staining is necessary to understand the structure of the tissue shown in Fig4f.

In Fig 4j the quantification graph does not reflect the western blot shown. The authors should also include statistical evaluation. The authors should better demonstrate the survivin-iRHOM2 protein interaction (Fig 4h) by doing surviving IP and western blot for iRHOM2.

Reply:

The statistical analysis in Fig 4j now Supplementary Fig.4j which includes additional experiments demonstrates increased apoptosis at 25mJ/cm² in TOC Sh iRHOM2 v TOC Sh Scr keratinocytes (p=0.0407).

As suggested by the reviewer we added the DAPI staining in the immunohistochemical images. The quality and resolution of the images presented in Fig.4e have also been improved. In addition, in Supplementary Fig.4f we added co-immunoprecipitation analysis showing that endogenous survivin is able to efficiently immunoprecipitate iRHOM2 in keratinocytes.

5. The data shown in Fig 5c is not clear, the authors should improve the staining. To demonstrate PLA specificity, the authors should also perform co-IP (Fig 5f).

Reply:

The Fig.5c corresponds to the qPCR for the anti-oxidant genes. We guess that the comment was referring to the Cygb staining (Fig.5d). The staining presented for Cygb in Fig.5d was not entirely clear and we added new staining and improved the quality and resolution of the images. In addition, we have now performed co-immunoprecipitation and observed that Cygb was able to immunoprecipitate iRHOM2 (Supplementary Fig.5g).

Reviewer #2 (Remarks to the Author):

Overall, the data reported by Arcidiacono et al support their conclusions that iRHOM2 is a novel p63 target gene. There are a few inconsistencies that the authors should address:

1. In Fig. 1e, iRHOM2 is expressed at high levels in HaCat cells have undergone Ca-induced differentiation, while Δ p63 is essentially absent under these conditions (day 5). What regulates the high levels of iRHOM2 in the absence of p63?
2. In Sup. Fig 3a, the authors report that PMA treatment of keratinocytes in vitro reduces p63 expression. What happens in vivo since PMA treated mouse skin exhibits a profound hyperproliferative phenotype?

3. This statement “Co-immunoprecipitation analysis showed that endogenous survivin was able to efficiently immunoprecipitate iRHOM2 in control keratinocytes (Fig. 4h).” is incorrect, iRHOM2 was able to efficiently immunoprecipitate surviving.

4. Since sulforaphane is a well-known inducer of Nrf2, which regulates numerous antioxidant response genes, can the authors exclude a role for Nrf2 in reduced ROS production? Are the same results obtained in Nrf2 knockout mice?

Reply:

We thank the reviewer and we have followed his/her constructive advice.

1. In Fig. 1e, iRHOM2 is expressed at high levels in HaCat cells have undergone Ca-induced differentiation, while p63 is essentially absent under these conditions (day 5). What regulates the high levels of iRHOM2 in the absence of p63?

Reply:

It has been proposed that p63 plays a dual role in keratinocyte differentiation, as it is required for initiating epithelial stratification (Koster et al., 2004).

Our data suggest that $\Delta Np63\alpha$, the major p63 isoform expressed in epidermis, is also the major p63 isoform that regulates iRHOM2 in keratinocytes. Our findings may appear to be in contrast with other studies. In fact, the majority of the $\Delta Np63$ target genes followed this principle, however it has been also reported that some genes do not respect this rule. For instance it has been described that ZNF750 is involved in epidermal differentiation, and shown to be upregulated during epidermal differentiation by $\Delta Np63\alpha$ which binds to its promoter (Sen et al., Dev Cell 2012). We have shown that during keratinocytes differentiation iRHOM2 expression is increasing at mRNA and protein levels similar to ZNF750. In normal skin, iRHOM2 is expressed in the cytoplasm and plasma membrane of the basal and suprabasal layers of the epidermis. As $\Delta Np63\alpha$ is essential to drive genes for the differentiation program, our data suggested that iRHOM2 is associated to this process.

2. In Sup. Fig 3a, the authors report that PMA treatment of keratinocytes in vitro reduces p63 expression. What happens in vivo since PMA treated mouse skin exhibits a profound hyperproliferative phenotype?

Reply:

PMA is a stressor inducer and activator of differentiation in primary keratinocytes (Hennings et al, JID 1987). Our results showed after PMA treatment a reduction of $\Delta Np63$ expression, in agreement with the observations in Nakanishi et al., JID 2006. We agree, in vivo, it has been reported that the treatment of mouse skin with PMA causes inflammation, enhances the thickness of the epidermis, and causes an increase in the expression of many differentiation markers. It has been described that PMA-treated skin exhibits several similarities with psoriatic skin, where the molecular differentiation control of p63 is severely disorganized (Marks et al., Environ Health Perspect 1993; Stanley et al., Skin Pharmacol 1991). It has been also reported dysregulation of p63 expression in psoriasis (Shen et al., The Journal of Dermatology 2005; Kim SY et al., J Cutan Pathol 2009). All together, these observations support an expected dysregulation of p63 expression in PMA treated mouse skin.

3. This statement “Co-immunoprecipitation analysis showed that endogenous survivin was able to efficiently immunoprecipitate iRHOM2 in control keratinocytes (Fig. 4h).” is incorrect, iRHOM2 was able to efficiently immunoprecipitate surviving.

Reply:

We agree and have amended the sentence. (Fig. 4f).

4. Since sulforaphane is a well-known inducer of Nrf2, which regulates numerous antioxidant response genes, can the authors exclude a role for Nrf2 in reduced ROS production? Are the same results obtained in Nrf2 knockout mice?

Reply:

Yes, we can exclude the role of Nrf2 in the reduction of ROS in TOC cells. We have added in Supplementary Figure 5c/d, the qPCR for NRF2 gene which showed no difference between control and TOC keratinocytes. In addition, no significant changes were found in keratinocytes depleted for iRHOM2.

It has been shown that in the skin, loss of Nrf2 or its functional inhibition prolonged the inflammatory response after wounding and enhanced the susceptibility to chemically induced carcinogenesis (Schafer et al., EMBO Mol Med 2012). In contrast, we have shown previously that iRHOM2^{-/-} mice displayed impaired shedding of ADAM17 substrates and abrogated the activation of inflammation (Maruthappu et al., Nat Com 2017).

Interestingly, the genetic activation of Nrf2 in keratinocytes of mouse skin severely affects the cornified envelope and impaired barrier function (Schafer et al., EMBO Mol Med 2012), while gain of function of RHBDF2 showed instead an improved barrier function in TOC skin (Brooke et al, Hum Mol Gen 2014). Taking together, these data suggest that Nrf2 and iRHOM2 function in distinct pathways.

Reviewer #3 (Remarks to the Author):

In the manuscript by Arcidiacono et al. the authors show a complex interplay between the master epidermal transcription factor p63 and iRHOM2 that has implications for the keratinocyte stress response. Perhaps more interestingly, there exists a differential interplay between p63 and iRHOM2 that is dependent on normal vs hyperproliferative states and the anatomical location of keratinocytes. The authors further demonstrate that the p63-iRHOM2 mediated signaling axis regulates ADAM17 activity and in turn influences downstream cellular functions such as proliferation, survival and oxidative defense. To extend their studies, the authors focus on additional players, which are direct p63 targets and/or binding partner of iRHOM2 - this include survivin and cytoglobin (CygB). Overall the manuscript is well-written and most of the conclusions are well-supported by experimental data. However, there are several issues that needs to be addressed to bolster the authors claim.

Reply:

We thank this reviewer for the constructive criticism and the helpful comments, which we have tried to fully address in this revised version of the manuscript as we detail point-by-point here below.

1) The p63 targets have been thoroughly identified by CHIP-seq experiments in both human and mouse keratinocytes. Yet, these published databases are not utilized for the identification of the specific targets that are chosen for the studies described in this paper. The examination of the proximal promoter region for the p63-binding sites is likely to be a rather limited and perhaps a misleading endeavour given that most of the p63 binding in the genome has been shown to be concentrated in the distal regulatory enhancers.

Reply:

We agree with the reviewer that the examination of the RHBDF2 proximal promoter region for the p63 binding sites is limited and to demonstrate that RHBDF2 is a p63 target gene required the validation with the available ChIPSeq data sets done in human and mouse keratinocytes (Kouwenhoven et al., PLoS genetics 2010; Sethi et al., Nucleic Acid Res 2016). We have now analysed the datasets and confirmed that the human and mouse p63 ChIP-seq sites are very well conserved for RHBDF2 (Fig.1a and Supplementary Fig.1a), ADAM17 (Fig.3g and Supplementary Fig.3b) and BIRC5 (Fig.4g and Supplementary Fig.4g). We have not included the previous analysis that was done with the promoter sequence analysed with the prediction software (genomatix.gsf.de) as, from the ChIPSeq data, p63 only weakly binds to the RHBDF2, ADAM17 and BIRC5 promoters.

2) The CHIP data needs to be validated by qPCR to show quantitative enrichment of p63 binding.

Reply:

We have now performed the ChIP-qPCR experiments and validated p63 binding to RHBDF2 (Fig.1b), ADAM17 (Fig.3h) and BIRC5 (Fig.4h) within the analysed region. Data showed that all three genes are direct p63 targets.

3) The p63 overexpression studies in the HEK293 cells (Fig 1) and its effects on endogenous iRHOM2 expression is not convincing.

Reply:

We understand the concern of the reviewer regarding the overexpression experiments. We have now repeated the transient transfection experiments and optimized the time of transfection for 9 hours of transactivation. This is in line with previous studies described for example in Candi et al., 2006 (JCS), where p63 isoforms could be detectable upon 3 hours post transfection. We observed that TAp63 α and Δ Np63 α significantly induce iRHOM2 expression at mRNA and protein levels (Fig.1d/e).

4) Since in both mouse and human keratinocytes, under most conditions, the Δ Np63 isoforms are overwhelmingly predominant, examination of TAp63 does not add much to the manuscript (Supp Fig 2 for e.g.)

Reply:

In accordance with the reviewer's suggestion we have now removed the figure.

5) The immunofluorescence studies of human and mouse skin samples will benefit from additional markers (K14 for e.g.) to better delineate the basal cells and the rest of the epidermis. Additionally, H&E stains will also allow better appreciation of the morphology in these skin sections.

Reply:

We have now included the H&E staining to appreciate the morphology of the human and mice skin in Fig.2a/d and also we have stained new sections for p63 and we added the DAPI images.

As suggested by the reviewer we have performed additional staining for K14 in human and mouse skin, to better delineate the basal cells of the epidermis in Supplementary Fig.2a/d. The quality and resolution of the images presented have also been improved.

6) The authors have used normal and TOC keratinocytes – yet the details about these cell lines are quite sparse. Presumably they are immortalized cell lines – what passages were used?

Reply:

As suggested by the reviewer, in materials and methods, we added more details about the cell lines used. TOC cells are immortalized keratinocytes from a Tylosis patient carrying the UK RHBDF2 mutation we have previously reported (Blaydon et al, Am J Hum Genet. 2012). Control keratinocytes were matching cells carrying the same immortalization with human papilloma virus (HPV-16) open reading frames E6 and E7. Control and TOC keratinocytes were used from passages 15 to 18.

7) What is the status of the epidermal differentiation program in the human TOC skin and in the iRHOM2-null mice? This aspect has been neglected in the current study.

Reply:

In Brooke et al., Hum Mol Gen 2014, we have shown that TOC-mutations in iRHOM2 are “gain of function” in nature and lead to an increase in ADAM17 dependent growth factor shedding and increased Transglutaminase 1 expression/activity. This increased level of EGFR ligand shedding observed in TOC keratinocytes suggests increased epidermal barrier function. Moreover, the phenotype observed in TOC epidermis and keratinocytes suggests the presence of a constitutive wound-healing-like state.

Our recent study (Maruthappu et al., Nat Com 2017), showed that TOC plantar skin exhibits marked thickening and hyperproliferative epidermis. Our studies reveal upregulation of both hyperproliferative markers (eg Keratin 16) and those of epidermal differentiation (for example TGM1). We also observed an upregulation of K14 expression, a component of the basal layer, as reported in this current work (Supplementary Fig.2a).

Overall, our data suggested that TOC skin epidermis displayed besides inflammation, the characteristics of hyperproliferation and abnormal differentiation of keratinocytes.

However, iRHOM2^{-/-} mice exhibit impaired maturation of ADAM17 and subsequent reduction in the shedding of several growth factors. We have shown that loss of ADAM17 shedding is associated with reduced TGM1 activity in the upper layers of the epidermis (Wolf et al., Sci Rep 2016).

In iRHOM2^{-/-} mice, we demonstrated that irhom2^{-/-} mice footpad reveal a thinner epidermis and reduced Ki67 and K16 (hyperproliferation markers) expression (Maruthappu et al., Nat Com 2017). Moreover, in this current study, we observed a reduction in the markers of the proliferative compartment with reduction in p63 and K14 expression.

Overall, our study indicate that gain of function mutation of iRHOM2 in human or loss of iRHOM2 in mice led to an abnormal differentiation program.

8) The authors have interesting data that led them to posit that “in interfollicular skin, iRHOM2 represses ΔNp63 expression, whilst in hyperproliferative footpad skin and in TOC keratinocytes, iRHOM2 positively regulates ΔNp63 expression”. There are several caveats to this assumption, principle being that these observations are merely correlative. In particular, the p63 immunofluorescence studies on the mouse samples as presented, does not really provide much support for the authors conclusions and lacks a much-required quantitative evaluation. Since this is really a novel finding, it would be worth-while for the authors to shore up these observations by independent experiments (Western blot of skin extracts). Also, the authors state that the irhom2^{-/-} mice footpad phenotype displays a thinner epidermis, the opposite of the phenotype observed in TOC palmoplantar epidermis. This is not easily discernible in the Fig.2D panel. Again, a representative histology panel will be useful.

Reply:

We have now included new sections in Fig.2d. In the new image p63 is clearly seen in the nuclei and we have included the DAPI staining. The quality and resolution of the images presented in Fig.2d have also been improved.

We also added the H&E staining to appreciate the morphology of the mice skin confirming our previous study (Maruthappu et al., Nat Com 2017) where we showed that iRHOM2^{-/-} mice footpad reveal a thinner epidermis whilst there was no significant difference in the thickness of their back skin. Effectively, our data showed that irhom2^{-/-} mice displayed increased p63 expression in the back skin but had reduced expression in their footpad epidermis compared to irhom2^{+/+} littermates. Additionally, we also validated these observations by qRT-PCR from mouse skin extracts in Fig.2e as well by Western blot analysis in Supplementary Fig.2b.

These observations for p63 expression were also confirmed in control and TOC keratinocytes depleted for iRHOM2 by Western blot analysis (Supplementary Fig.2c).

9) Statistical analysis is not adequate.

Reply:

All experiments were performed at least three times independently or more indicated in the legends. The data are expressed as the mean ± standard error of mean (SEM).

All statistical analyses were performed with Prism 6 Software. Data were analysed by the unpaired/paired two-tailed Student's t test and one way ANOVA Dunnett's multiple comparison test. [p < 0.05 (), p < 0.01 (**), p < 0.001 (***)]*

10) Finally, the manuscript as it stands now seems more of an attempt to bring together a few players that fit into the p63-iRHOM2 axis but falls short in breaking new ground in

terms of either p63 or iRHOM2 biology. The limited studies performed on the mouse model of irhom2KO adds further confusion to the story line and the authors do not address the mechanistic basis of the footpad phenotype and its relationship to TOC states.

Reply:

The palmoplantar epidermis mirrors many aspects of hyperproliferative skin conditions including TOC such as expression of stress/wound healing proteins like keratin 16 (Maruthappu et al., Nat Com 2017). Thus combining these two model systems (mouse paw and human TOC skin) is providing mechanistic insight into how the keratinocyte responds to both physiological and disease-associated stress. Here, this study shows different signalling mechanisms between normal interfollicular and palmoplantar epidermis and this naturally physiologically stressed palmoplantar epidermis mirrors disorders of keratinocyte hyperproliferation. Furthermore we demonstrate iRHOM2 and p63 as key players in this differential stress associated signalling. We would like to highlight the key points of the paper.

We have shown that:

- i) p63 regulates the iRHOM2-ADAM17 axis in keratinocytes.
- ii) There is a reciprocal regulation between p63 and iRHOM2, and indeed with ADAM17.
- iii) This in turn regulates p63 target genes implicated in inflammation, apoptosis and oxidative stress.

Reviewer #1 (Remarks to the Author):

Arcidiacono et al. have performed several constructive experiments, significantly improving the manuscript. Overall, the work expands the knowledge on p63 biology. In particular they conclude that iRHOM2 seems to be a target for hyperproliferation and inflammatory skin diseases. Despite the new data, still additional experiments are required for improving some of the images reported.

1. Figure 1. Quality is now improved. ChipSeq has been compared to human and mice keratinocytes, confirm the previously reported data. Also the luciferase assay is fine. However, this reviewer is still not happy with the quality of the images in figure 2 a,b. A better confocal quality is required.
2. Figures 3 and 4. The new data are fine, with the exception of the IF image in figure 4e, that requires higher quality.
3. Figure 2. See above, A better quality image is still required
4. Figure 4. This is now fine (except image in figure 4e, as above).
5. Figure 5c is not fine.

Reviewer #2 (Remarks to the Author):

The authors have successfully addressed my concerns

Reviewer #3 (Remarks to the Author):

The authors have satisfactorily responded to the reviewers comments and suggestions, and as a result the manuscript is significantly improved.

Reviewer #1 (Remarks to the Author):

Arcidiacono et al. have performed several constructive experiments, significantly improving the manuscript. Overall, the work expands the knowledge on p63 biology. In particular they conclude that iRHOM2 seems to be a target for hyperproliferation and inflammatory skin diseases.

Despite the new data, still additional experiments are required for improving some of the images reported.

1. Figure 1. Quality is now improved. ChipSeq has been compared to human and mice keratinocytes, confirm the previously reported data. Also the luciferase assay is fine. However, this reviewer is still not happy with the quality of the images in figure 2 a,b. A better confocal quality is required.

Response: *Compared to our previous submission, the quality and resolution of the images presented in Fig. 2a and Fig.2b have been improved. The confocal pictures have been taken with a resolution of 1024x1024 and imported at 600dpi. However when we converted the data in PDF, the images did not show the quality expected, probably because they were compressed and showed loss of resolution during the pdf creation. Now we have saved the pictures in another format that saved the resolution when we converted them in PDF.*

2. Figures 3 and 4. The new data are fine, with the exception of the IF image in figure 4e, that requires higher quality.

Response: *For the reasons explained above, we have saved the pictures in another format that saved the resolution when we converted them in PDF.*

3. Figure 2. See above, A better quality image is still required

Response: *The answer is detailed in point 1.*

4. Figure 4. This is now fine (except image in figure 4e, as above).

Response: *For the reasons explained above, we have saved the pictures in another format that saved the resolution when we converted them in PDF.*

5. Figure 5c is not fine.

Response: *The reviewer was not clear about what was required for Fig.5c, so our guess is that maybe the representation of the data was repetitive with all the controls. The controls are now represented by a dashed line as explained in the figure legend.*